# Sequence Stratigraphy and Geochemistry of Oil Shale Deposits in the Upper Cretaceous Qingshankou Formation of the Songliao Basin, NE China: Implications for the Geological Optimization of In Situ Oil Shale Conversion Processing

**Penglin Zhang [1,2], Yinbo Xu [1,*], Qingtao Meng [1,2,*], Zhaojun Liu [2], Jiaqiang Zhang [1], Lin Shen [2] and Shuaihua Zhang [2]**

1    The Key Laboratory of Unconventional Oil and Gas Geology, China Geological Survey, Oil and Gas Survey, China Geological Survey, Beijing 100083, China; zhangpl320@sina.cn (P.Z.); zjq1965@sohu.com (J.Z.)
2    Key Laboratory for Oil Shale and Paragenetic Minerals of Jilin Province, College of Earth Sciences, Jilin University, Changchun 130000, China; liuzj@jlu.edu.cn (Z.L.); shenlin17@mails.jlu.edu.cn (L.S.); zhangsh18@mails.jlu.edu.cn (S.Z.)
*    Correspondence: xuyinbo87@126.com (Y.X.); mengqt@jlu.edu.cn (Q.M.)

**Abstract:** The Songliao Basin contains some of the largest volumes of oil shales in China; however, these energy sources are located in areas covered by arable land, meaning that the best way of exploiting them is likely to be environmentally friendly in situ conversion processing (ICP). Whether the oil shales of the Songliao Basin in the Qingshankou Formation are suitable for ICP remain controversial. In this paper, through sequence stratigraphic correlations, three main thick oil shale layers (N1, N2, and N3) of the Sequence1 (Sq1) unit in the first member of Qingshankou Formation ($K_2qn^1$) are confirmed as consistently present throughout the Southeastern Uplift region of the basin. The spectral trend attributes reflect that the lake reached a maximum flood surface of the $K_2qn^1$ in N2 oil shale layer, and the total organic carbon (TOC) and Fischer assay (FA) oil yield are significantly increasing. The N2 and N3 oil shale layers were deposited in a high lake level environment associated with ingressions of ocean water. The oil shale in these layers with the characteristics of high TOC (maximum of 23.9 wt %; average of 7.2 wt %), abundance of aquatic organic matter (OM) (maximum hydrogen index (HI) of 1080.2 mg/g; average of 889.9 mg/g) and carbonate contents (maximum of 29.5%; average of 15.4%). The N2 and N3 oil shale layers have higher brittleness index (BI) values (generally 40–50%), larger cumulative thicknesses (maximum of 13.3 m; average of 12.0 m), and much higher source potential index (SPI) values (0.92 and 0.88 tHC/m², respectively) than the N1 oil shale layer within Sq1 transgressive system tracts (TST), indicating that the N2 and N3 layers are prospective targets for ICP. In addition, oil shales buried to depths of <1000 m have strong hydrocarbon generation capacities that make them suitable for ICP.

**Keywords:** sequence stratigraphy; hydrocarbon generation potential; rock brittleness; insitu conversion processing; Songliao Basin

## 1. Introduction

The future of domestic oil supplies in China is currently uncertain as a result of a lack of discovery of high-quality oil reserves and difficulties in maintaining stable hydrocarbon production despite the fact that China has vast oil shale resources [1]. Chinese oil shale reserves are estimated to be 978 billion tons of oil and 61 billion tons of shale oil, of which recoverable reserves are 61 billion tons of oil by

the traditional distillation method. The Upper Cretaceous of the Songliao Basin represents the most important known resource in China [2,3]. The Songliao Basin of northeastern China crops out over an area of $26 \times 10^4$ km$^2$ and is covered over $559 \times 10^4$ hectares of farmland [3]. The basin recorded a rapid and large-scale lake transgression event during the deposition of the first member of the Qingshankou Formation (K$_2$qn$^1$). Thick and dark shales and oil shales rich in organic matter are both widely distributed throughout the K$_2$qn$^1$ stratum in the basin and are thought to be ideal targets for oil extraction. However, although the K$_2$qn$^1$ member has been the focus of a significant amount of research and represents a highly prospective oil shale resource, the cultivated land use in this area has restricted the development of these oil shale deposits.

A significant amount of recent research has focused on the development of environmentally friendly and effective technologies and approaches for in situ conversion processing (ICP) [1,4,5]. The in situ conversion processing (ICP) of oil shale underground at the depth ranging from 300 to 3000 m is a physical and chemical process caused by using horizontal drilling and electric heating technology, which converts heavy oil, bitumen, and various organic matter into light oil and gas in a large scale, which can be called "underground refinery" [1]. Research into the ICP of oil shale in China has focused on the Songliao Basin of Jilin Province, where United Strength Power Holdings Ltd. undertook a pilot test of the in situ conversion processing and chemical distillation of oil shales in this basin. This test initially yielded 5.2 tons of oil, with a further 8.86 tons of oil generated during the middle part of this test [1]. Further research was undertaken in 2015 by Jilin University in cooperation with Israel Asia technology co., LTD, leading to leading to a pilot test of the ICP of oil shales using a topochemical reaction (TS-A) approach, with this pilot test successfully generating crude oil [1,6–8].

Previous research has indicated that the oil yield during the ICP of oil shales is affected by the oil shale layer thicknesses and brittleness, and the abundance, type, and maturity of organic matter (OM). Therefore, this study focuses on the distribution of these oil shales within the sequence stratigraphic framework of the basin, the mineralogy of the oil shales, and the organic geochemical characteristics of the OM within these units, all of which are key for successful ICP [4–8]. The sequence stratigraphy of the study area can determine the distribution of source rocks by analyzing eustatic lake level changes with depth [9,10], providing a guide for future exploration. However, K$_2$qn$^1$ are dominated by dark mudstone, and the traditional sequence division scheme is significantly influenced by subjective human factors. The sequence boundary of K$_2$qn$^1$ is controversial and poorly correlated; it is also unable to be used to effectively predict the distribution of oil shales and plays a limited role in guiding exploration and production. In order to minimize the impact of human factors, make the sequence division scheme more objective, and highlight the prediction of sequence stratigraphy on oil shales, the sequence of K$_2$qn$^1$ was divided based on integrated prediction error filter analysis (INPEFA), coupled with drilling data in this paper [11]. This study focuses on the sequence stratigraphy, mineralogical and geochemical characteristics of the first member of the Qingshankou Formation in the Songliao Basin, and it uses the resulting data to identify suitable oil shale layers for future oil production by ICP.

## 2. Geological Setting

The Songliao Basin is a large composite continental basin in northeastern China that contains abundant oil and gas resources. The basin contains Jurassic fault-bound and post-Jurassic rhomboidal depressions, all of which strike NE–SW. The basin is ca. 750 km long and 350 km wide and covers an area of 260,000 km$^2$. It is divided into six first-order tectonic units, which are the Central Depression, Northern Slope, Western Slope, Northeastern Uplift, Southeastern Uplift, and Southwestern Uplift [12,13]. The upper parts of the basin are dominated by Mesozoic–Cenozoic sediments (Figure 1), including voluminous, thick, and widespread source rocks of the Upper Cretaceous Qingshankou (K$_2$qn) and Nenjiang (K$_2$n) formations [11].

This study focuses on the Qingshankou Formation (K$_2$qn), which includes a first member (K$_2$qn$^1$) that records rapid and large-scale lake transgressions that deposited thick, dark shale units and oil shales rich in organic matter throughout the basin. The second (K$_2$qn$^2$) and third (K$_2$qn$^3$) members of

the Qingshankou Formation record a basin-wide regression and a decrease in the area of the associated lake, with northern, western, and southern fluvial systems migrating toward former lake areas and depositing more sand bodies [14,15].

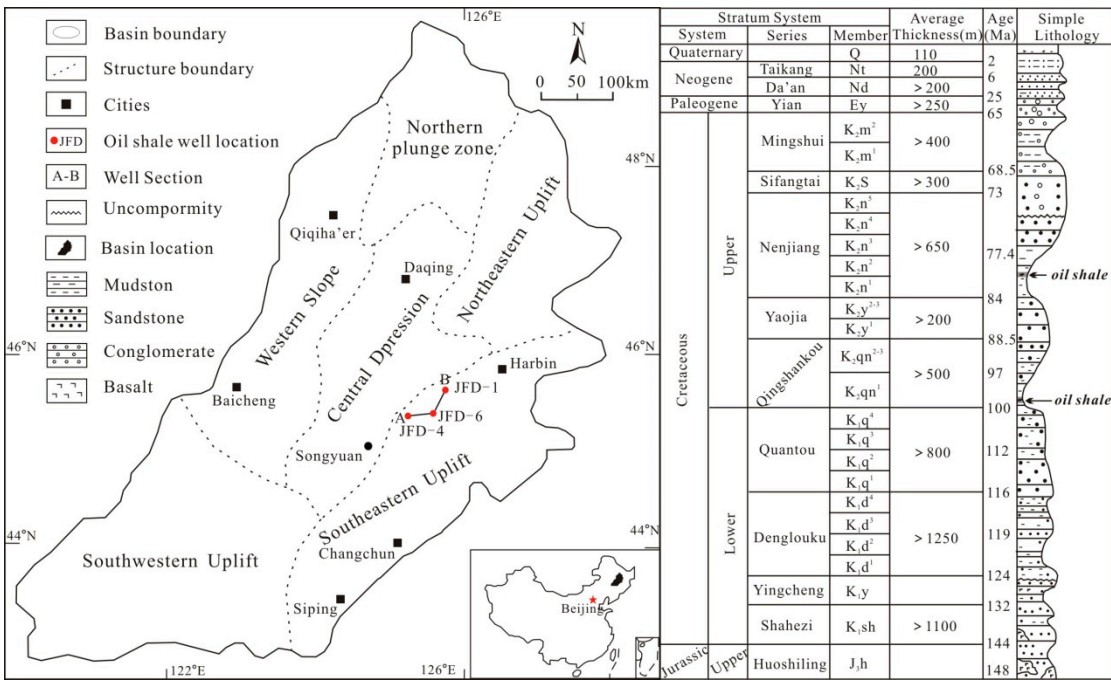

**Figure 1.** Map and stratigraphic column showing the location, major structural units, drilled well sections, and stratigraphy of the Songliao Basin [15].

## 3. Samples and Methods

### 3.1. Materials and Sampling Location

Samples from the Songliao Basin were collected from three wells in the Southeastern Uplift zone, all of which were drilled by the Geological Survey of China, which are wells JFD-1 (sampling depth of 330–450 m), JFD-4 (sampling depth of 700–740 m), and JFD-6 (sampling depth of 1030–1090 m). Well JFD-1 contains the thickest high quality oil shale layers (Figure 2a). All Fischer assay (FA) oil yield samples were taken at 1 m intervals. A subset of these samples was used to determine total organic carbon (TOC) contents and was used for Rock–Eval pyrolysis, organic geochemical analysis, X-ray diffraction analysis, as well as the determination of maceral compositions. The majority of the samples are black–gray, dark gray, and brown–gray oil shales or gray to light gray mudstones.

### 3.2. Spectral Trend Attribute Analysis and Sequence Stratigraphic Division

Spectral trend attribute analysis is a stratigraphic cycle identification technique that uses well log spectral data [16]. This approach is based on a cyclical stratigraphic theoretical basis and converts well logs into a single integrated prediction error filter analysis (INPEFA) curve using modern digital signal processing techniques. INPEFA highlights the often hidden characteristics of stratigraphic cycles within well logs [17]. This enables the in-depth analysis of sequence stratigraphic and sedimentary cycles. Previous research indicates that the $K_2qn^1$ member of the Songliao Basin is dominated by open lake facies shale units, suggesting that lowstand systems tracts (LST) are absent within the third-order sequence of the $K_2qn^1$ member [18]. However, significant changes in eustatic lake level during the deposition of the $K_2qn^1$ member caused significant variations in the abundance of shale lithologies. Therefore, the changes in the amplitude of the Gamma Ray (GR) curve are likely to indicate changes in sedimentary cycles. In addition, the GR curve is only slightly affected by borehole conditions and

is already included within the logging suites of existing wells in this area. Thus, these data were converted to yield INPEFA curves that, in turn, were used to define the sequence stratigraphy using the following approaches.

The first step in obtaining INPEFA curves was to analyze the GR curve data using maximum entropy spectral analysis (MESA), which yielded a MESA curve that was then analyzed using prediction error filter analysis (PEFA). This generated the numerical error between predicted MESA values and measured well log values at the corresponding depth (i.e., a PEFA curve), where the PEFA values are the result of subtracting the actual data value from the filtered value. The PEFA curve is an irregular dentate curve that varies vertically and provides an indication of stratigraphic continuity: negative peaks represent possible flooding surfaces, positive peaks represent possible sequence boundaries, and different peaks are indicative of isochronous surfaces within difference stratigraphic sequences [16–18]. The analysis of the cycles within the $K_2qn^1$ member indicates that the PEFA curve obtained from the GR curve after filtering yields good results in this area. Finally, integrating processing was performed on the PEFA curve [16] to yield a final INPEFA curve that was used for sequence division, based on the presence of positive and negative trends and the transitions between these areas of the curve. Sedimentary sequences with overall negative trends (i.e., INPEFA curve values that gradually decrease from bottom to top) are indicative of climates that gradually become more arid and of regressive sedimentary sequences, whereas overall positive trends (INPEFA curve values that gradually increase from bottom to top) are indicative of transgressive sedimentary sequences [16,18]. Changes from positive to negative or vice versa are indicative of possible sequence boundaries or internal interfaces within individual parts of the sequence. Negative turning points within a given interval also indicate the minimum depositional rate within that interval and may represent flooding surfaces, whereas positive turning points may represent sequence boundaries [16,18].

### 3.3. X-Ray Diffraction Analysis

The mineralogy of a total of 36 cuttings from three research wells in the study area, the majority of which were oil shales, was determined by whole-rock and clay mineral (<2 μm) X-ray diffraction (XRD) analysis. XRD mineralogical patterns were obtained using a Bruker XRD diffractometer with CuK$\alpha$ radiation operated at 40 kV and 40 mA at a step size of 0.025°/s. This analysis used the sample preparation and mineral identification approaches outlined in Brindley and Brown [19] and Moore and Reynolds [20]. Prior to analysis, samples were gently powdered in an agate mortar and were packed in cavity mounts for bulk mineral analysis. Whole-rock powder samples were scanned over a 2° to 70° 2θ range. Clay minerals were extracted by dispersing samples in distilled water and the subsequent separation of the <2 μm clay-sized fraction from the suspension by programmed centrifugation without any pre-chemical treatment. Oriented <2 μm clay-sized fraction specimens were prepared by sedimenting the extracted suspension onto glass slides and drying at room temperature. The <2 μm slides were from 2° to 30° 2θ in an air-dried (AD) state after ethylene glycol treatment (EG) and heating to 550 °C for 4 h. Peak indexing, mineral identification, and background stripping were all performed using the X'pert High Score software package. It should be pointed out that in the process of X-ray diffraction, the directional arrangement of mineral crystals may affect the analysis results [20]. Therefore, when we analyze the mineral composition through XRD, we combine the thin section observation.

### 3.4. Organic Geochemical Analysis

Total organic carbon (TOC) and Rock–Eval pyrolysis analyses were undertaken on a total of 92 oil shale and mudstone samples. The TOC analyses used 1 g of a powdered sample that was treated with dilute hydrochloric acid (HCl) to remove carbonates before TOC values were determined using a LECO-CS 244 instrument. The resulting TOC values were recorded in weight percent (wt %) values relative to the initial rock sample. Rock–Eval pyrolysis used ca. 50 mg of a bulk powdered sample that was analyzed using a GEO-IMT-2005 Rock–Eval pyrolysis instrument to determine the characteristics

of the generated hydrocarbons and the type of organic matter (i.e., kerogen). Measured parameters during this pyrolysis include the amount of hydrocarbons that were generated but not expelled from the source rock (S1), the remaining hydrocarbon generation potential (S2), and the maximum temperature of hydrocarbon generation ($T_{max}$). The hydrogen index (HI) values reflecting the quantity of unoxidized hydrogen in the system and pyrolysis hydrocarbon index (HCI = $S_1$/TOC in mg HC/g TOC terms) values were also calculated during this analysis. Four oil shale samples were also used for vitrinite reflectance (Ro) analysis using a Leica DMRX microscope equipped with a photometer. This analysis used mean particle reflectivity values that were recorded in percentage units.

### 3.5. Organic Petrography

Microscopy was undertaken on four oil shale and one mudstone sample during this study using polished whole-rock blocks cut perpendicular to bedding planes. This analysis used a Leica MPV microscope and oil immersion objectives, with maceral analysis performed using reflected white and fluorescent light and a 50× objective lens. A total of 500 points per polished block were counted using a single-scan method, and vitrinite reflectance values were measured using a 100× objective lens under plane-polarized light at a wavelength of 546 nm.

## 4. Results

### 4.1. Lithostratigraphy of $K_2qn^1$

The $K_2qn^1$ member is dominated by gray–black mudstone, dark gray oil shale, and locally developed dolomite units. The lower part of the member within the Southeastern Uplift part of the basin contains three thick oil shale layers (N1, N2, and N3) that are widespread throughout this area. The N2 and N3 oil shale layers have generally high FA oil yield values (averages of 5.8% and 5.0%, respectively; Figure 2a–c) and both layers are thick (average of 6.0 and 6.2 m, respectively; Figure 2a–c), but the N1 oil shale layer has a low FA oil yield (average of 4.1%) and is much thinner (average of 2.1 m; Figure 2a–c).

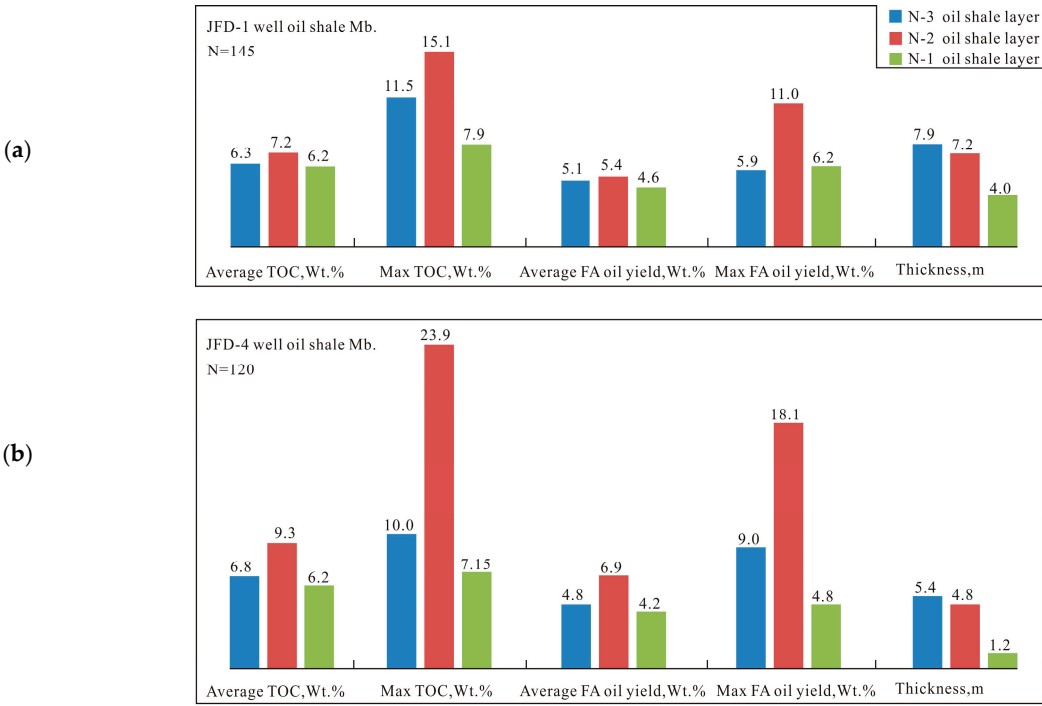

**Figure 2.** *Cont.*

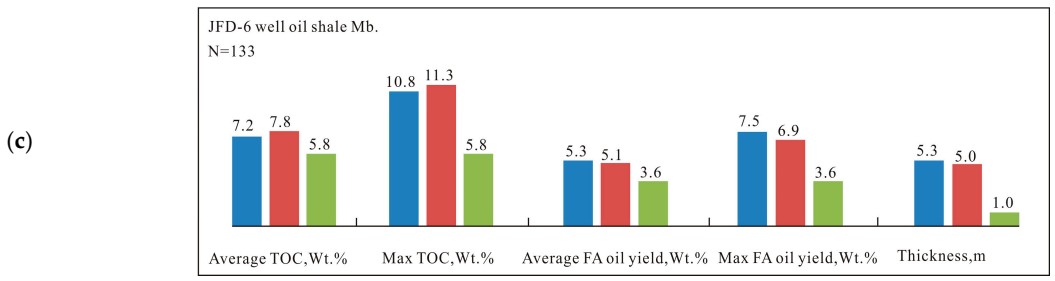

**Figure 2.** Histograms showing variations in the total organic carbon (TOC) and Fischer assay (FA) values and thicknesses of the (**a**) JFD-1, (**b**) JFD-4, and (**c**) JFD-6 oil shale layers.

## 4.2. Whole-Rock Powder Analysis

A powdered sample in an agate mortar scanned over a 2° to 70° 2θ range by XRD, determined that quartz, albite (plagioclase), dolomite, calcite, analcite, and pyrite are the main non-clay minerals within the analyzed samples (Figures 3 and 4). These samples have clay mineral components that contain smectite, illite, chlorite, kaolinite, and mixed interlayered illite/smectite (I/S) (Figures 3 and 4). Quartz is the most abundant non-clay mineral, and it is present in all samples (Figure 4). The majority of feldspar within these samples is albite, although rare K-feldspar is also present. Calcite and dolomite are present in trace amounts in most samples, but they are very rare in low TOC samples. Pyrite is also present in trace amounts in most shales, indicating that the $K_2qn^1$ shales were deposited in a relatively reduced environment. These mineralogical data did not indicate the presence of any systematic changes in the abundance of both clay and non-clay minerals with burial depth.

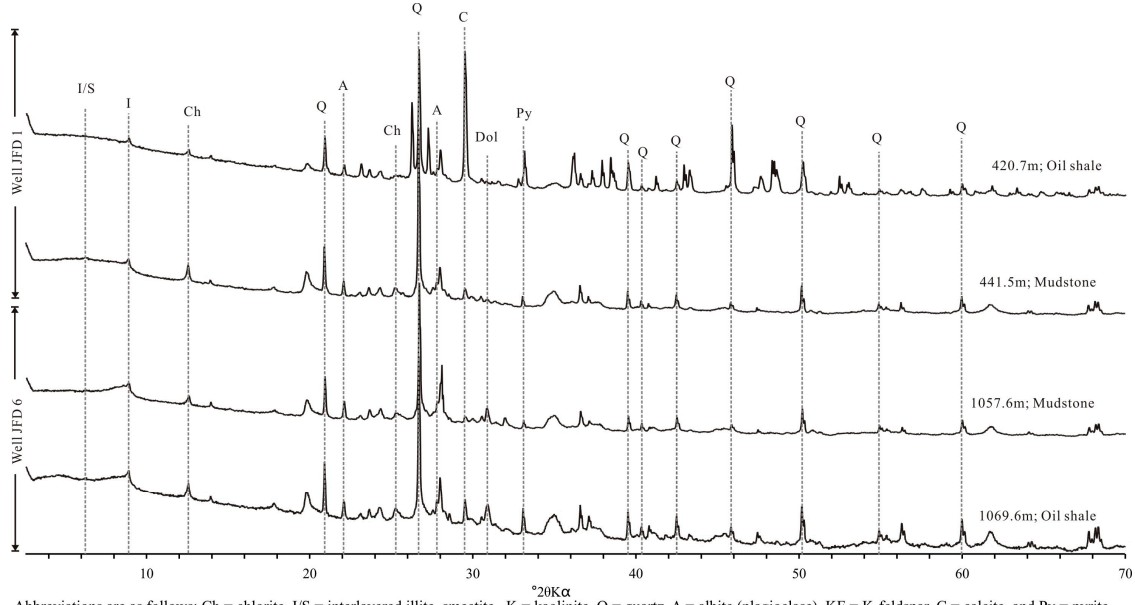

Abbreviations are as follows: Ch = chlorite, I/S = interlayered illite–smectite, K = kaolinite, Q = quartz, A = albite (plagioclase), KF = K-feldspar, C = calcite, and Py = pyrite.

**Figure 3.** Representative whole-rock X-ray diffraction patterns for samples from the $K_2qn^1$ oil shale layers in the southeastern Songliao Basin.

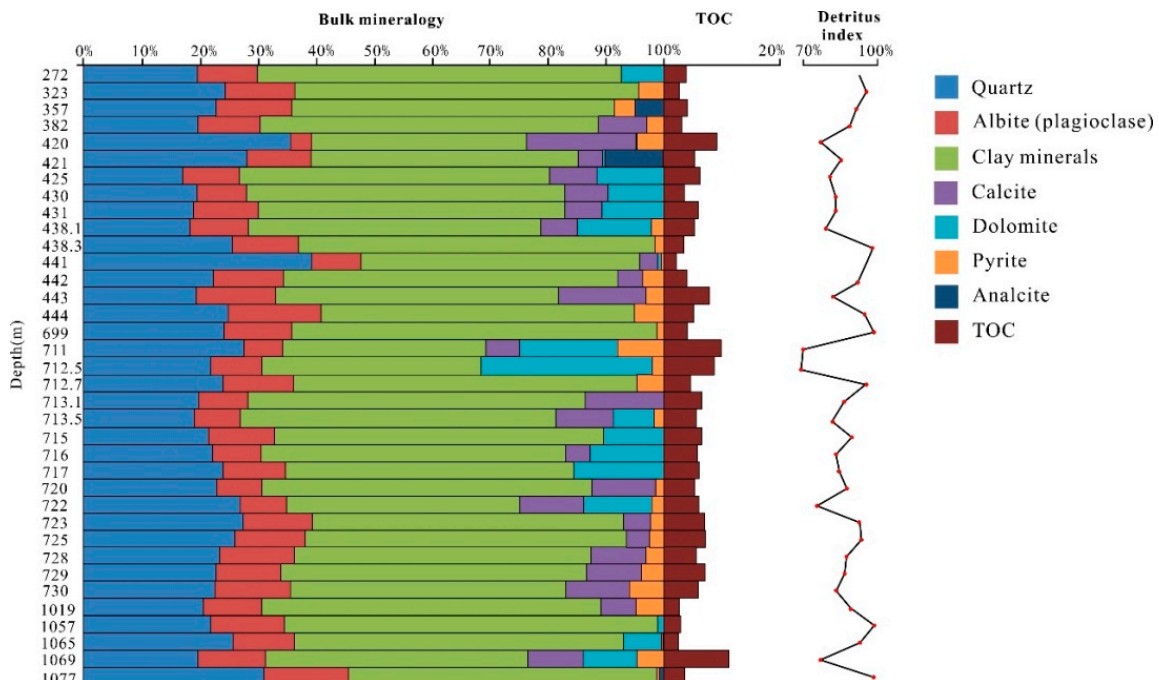

**Figure 4.** Chart showing variations in mineralogy and TOC values with depth in the southeastern Songliao Basin.

### 4.3. Clay Mineralogy (<2 μm Fraction)

The samples from the study area have clay mineral assemblages of smectite, illite, kaolinite, chlorite, and I/S (Figures 5 and 6). Chlorite, illite, kaolinite, and I/S are present in all samples, whereas smectite is generally only present in shallowly buried samples (Figures 5 and 6). Chlorite and illite abundances also positively correlate with increasing depth, whereas I/S abundances decrease with depth. There is no obvious relationship between kaolinite abundance and burial depth.

Discrete (i.e., well crystallized) illite can be identified by the presence of 9.96, 5.0, and 3.3 Å peaks that correspond to the 001, 002, and 003 reflections, respectively, within AD data, with these peaks remaining relatively constant for AD and EG treated samples (Figures 5 and 6). Smectite, kaolinite, and chlorite are identified by their characteristic peaks at 17.0 Å for smectite, 7.14 and 3.57 Å for kaolinite, and 14.2, 7.1, 4.71, and 3.53 Å forchlorite [21] (Figure 5). The I/S minerals are identified by their asymmetrical, non-harmonic, and low-angle broad peaks that shift from 10.77 to 13.8 Å in AD data to 11.12–17.2 Å in EG data. The intensity of the I/S peaks generally decreases with increasing burial depth, whereas the intensity of the smectite and illite peaks both increase with increasing depth (Figures 5 and 6).

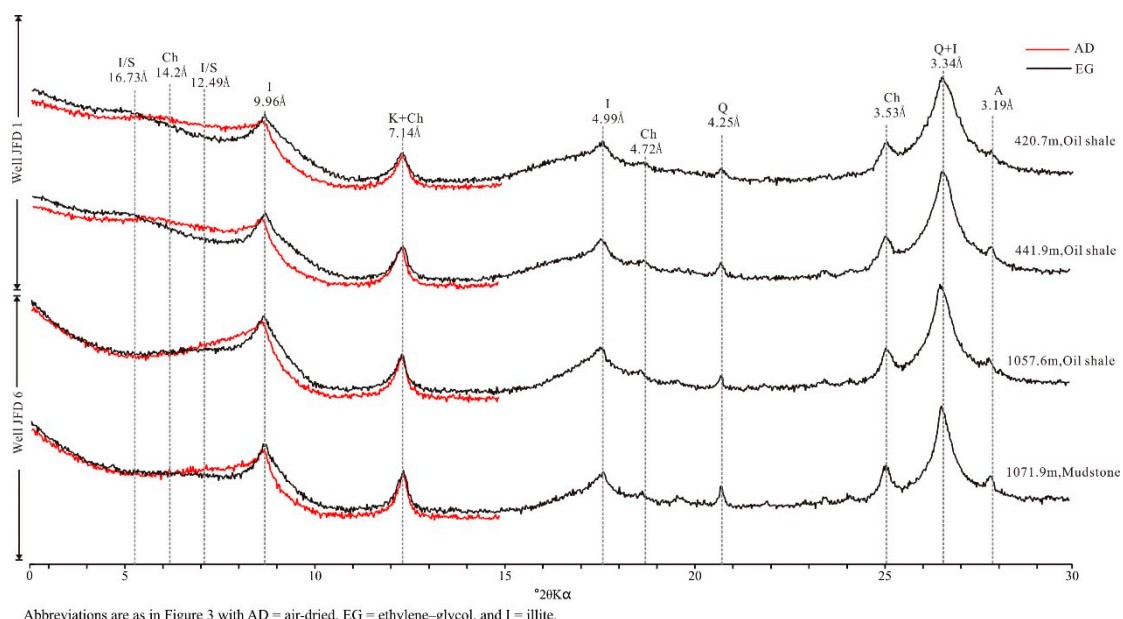

**Figure 5.** Representative X-ray diffraction patterns for oriented <2 μm clay fractions from the $K_2qn^1$ oil shale layers of the southeastern Songliao Basin.

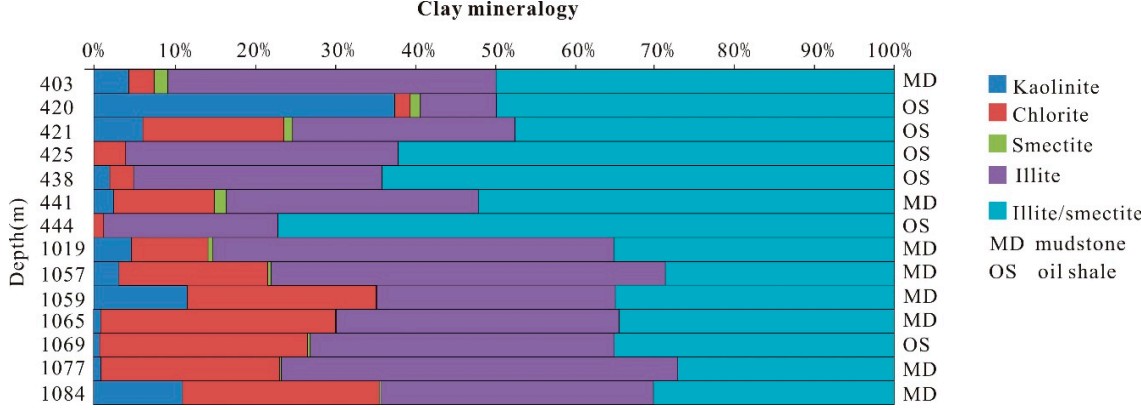

**Figure 6.** Diagram showing variations in clay mineralogy with depth in the $K_2qn^1$ oil shales of the southeastern Songliao Basin.

### 4.4. Bulk Geochemical Data

The N2 oil shale layers have TOC values of 5.0–23.9 wt % (average of 7.8 wt %) that are higher than the TOC values of the N1 (maximum of 7.9 wt %; average of 6.3 wt %) and N3 (maximum of 11.5 wt %; average of 6.7 wt %) oil shale layers. The mudstone samples from the oil shale intervals in the study area have an average TOC value of 2.9 wt %. A relatively high content of total sulfur (TS) occurs in $K_2qn^1$, arguing for relatively high alkalinity [22]. The N2 and N3 oil shale layers have higher TS contents (average of 0.77 wt %) than the N1 oil shale layer (average of 0.72 wt %) and the mudstones in this area (average of 0.60 wt %).

The N1 oil shale layer has HI (S2×100/TOC) values of 614–737 mg HC/g TOC and contains type-I and type-II kerogens (Figure 7) [23]. In comparison, the N2 and N3 oil shale layers contain type-I kerogen with average HI values of 927 and 852 mg HC/g TOC, respectively. The average "true" HI values of the samples from the $K_2qn^1$ member were determined using a regression line defined by $S_2$ data plotted against TOC values, which yielded very variable values [24]. The average "true" HI values for the N3 and N2 oil shale layers are 1105 and 959 mg HC/g TOC, respectively ($r^2 = 0.92$ for N1 and 0.98 for N2). However, the N1 oil shale layer yielded a lower "true" HI value of 760 mg HC/g TOC

and with a poorer correlation between the $S_2$ and TOC values for these samples ($r^2 = 0.56$). These data indicate that the N2 and N3 oil shale layers contain type-I kerogen, whereas the N1 oil shale layer contains type-I and type-II kerogens (Figure 7).

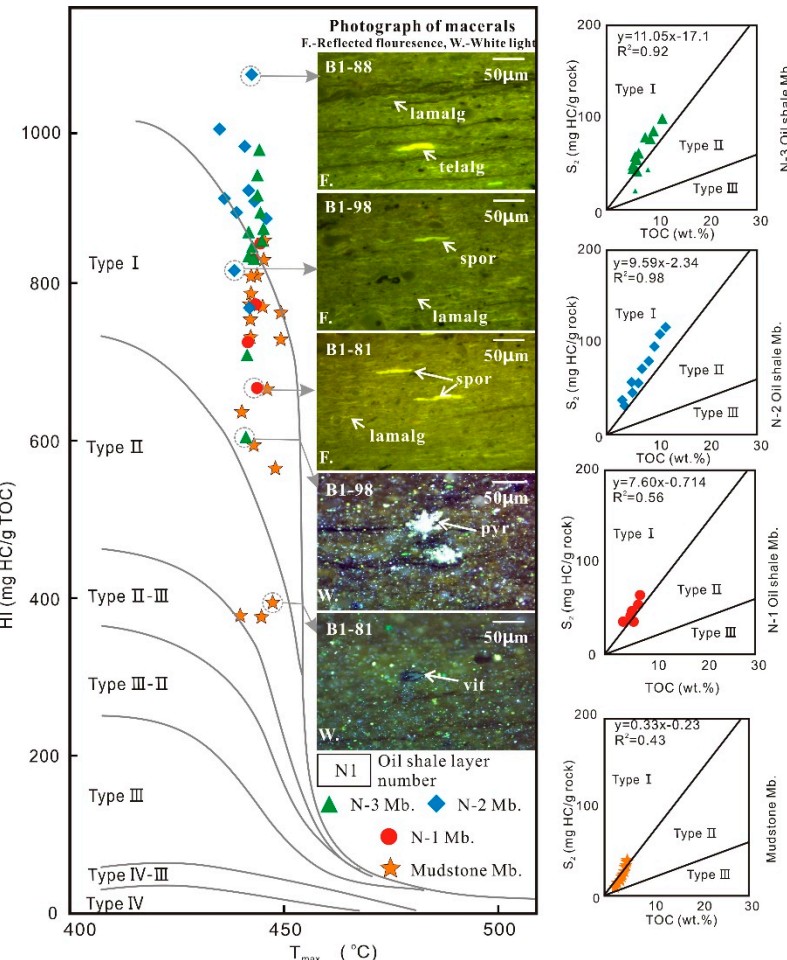

**Figure 7.** Discriminant map[18] showing variations in the types and abundances of kerogen in oil shales of the $K_2qn^1$ member with abbreviations as follows: Lamalg = lamalginite, Telalg = telalginite, Spor = sporinite, Vit = vitrinite, and Pyr = pyrite.

The samples analyzed during this study have $T_{max}$ values that range from 435 °C at a depth of 433 m to 447 °C at 1061 m. The oil shale samples from the shallowest JFD-1 well have $T_{max}$ values of 435–446 °C that reflect immature and low maturity conditions [25,26], whereas all of the oil shale samples in the other two wells from depths of 699–1079 m and with $T_{max}$ values > 437 °C are classified as low maturity.

### 4.5. Sequence Stratigraphic Features of the $K_2qn^1$ Member

Based on the spectral trend attribute analysis of the GR curve in a typical single well (JFD-1), spectral trend attribute analysis was performed on the GR curves of 3 wells. Then, the third-order sequence stratigraphic framework of the $K_2qn^1$ in the Songliao Basin was established by referring to the five sequence stratigraphic framework profiles throughout the entire region. Comparing the GR curves with this framework indicates that the spectral trends of these curves correlate well with variations in the stratigraphy and sedimentary cyclicity of the $K_2qn^1$ member in the Southeastern Uplift zone. In $K_2qn^1$, boundaries of the three third-order sequences and their internal constitutive characteristics are clearly evident in Figure 8.

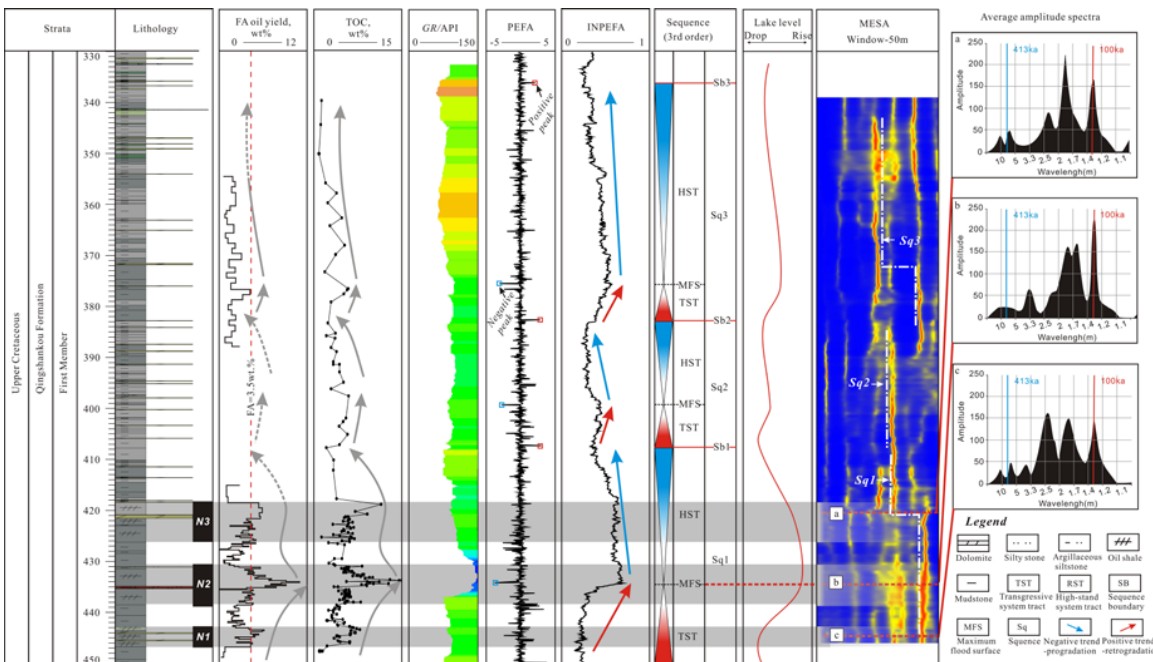

**Figure 8.** Stratigraphic column showing locations of sequence stratigraphic divisions and variations in other key characteristics with depth in the first member of the Qingshankou Formation ($K_2qn^1$) member encountered in well JFD-1 within the Songliao Basin.

Sequence Sq1 includes the three oil shale layers and contains transgressive system tracts (TST) in the N1 oil shale layer and lower part of the N2 oil shale layer, whereas highstand system tracts (HST) are generally located within the N3 oil shale layer and the upper part of the N2 oil shale layer. The maximum flooding surface (MFS) correlates to oil shales with the highest oil yield, and this surface has a prominent well logging response feature (Figure 8) in the form: (1) a rapid decrease in GR curve data; (2) a low wavelength but high amplitude MESA data where amplitude peaks occur at 100 ka intervals, indicating that these short-term cycles are predominantly controlled by eccentricity variations [16]; (3) negative peaks in PEFA curve data; and (4) INPEFA curve values close to one that represent a negative turning point from positive to negative trends.

The TST are dominated by dark gray to gray–black mudstones and oil shales, all of which become darker and contain more carbonate minerals from the base to the top of these tracts (Figure 9). The oil shales in this unit with the characteristic of bioclastic bands (Figure 9). Thin sections of the cores from these intervals are dominated by preferentially oriented bioclastic and detrital mineral bands. These intervals also correlate with amplitude highs on the GR curves and positive trends in the INPEFA, FA oil yield, and TOC curves. In comparison, the HST are dominated by dark black oil shales and mudstones that become lighter in color and have FA oil yield and TOC values that decrease from base to top (Figure 9). The top of the HST is associated with minimum GR curve values, significant decreases in FA oil yields, and negative INPEFA trends (Figure 8). The Sq2 sequence is dominated by dark gray and gray mudstones with HST intervals containing thin (<1 m) oil shale layers where INPEFA curves correlate with TOC values (Figure 8). In comparison, the Sq3 sequence is dominated by gray and dark green mudstones that have generally low TOC values. In summary, these sediments record a rapid increase in lake level during the deposition of the N2 oil shale layer before the lake reached a maximum flood surface and then dropped slowly over time, albeit with fluctuating lake levels.

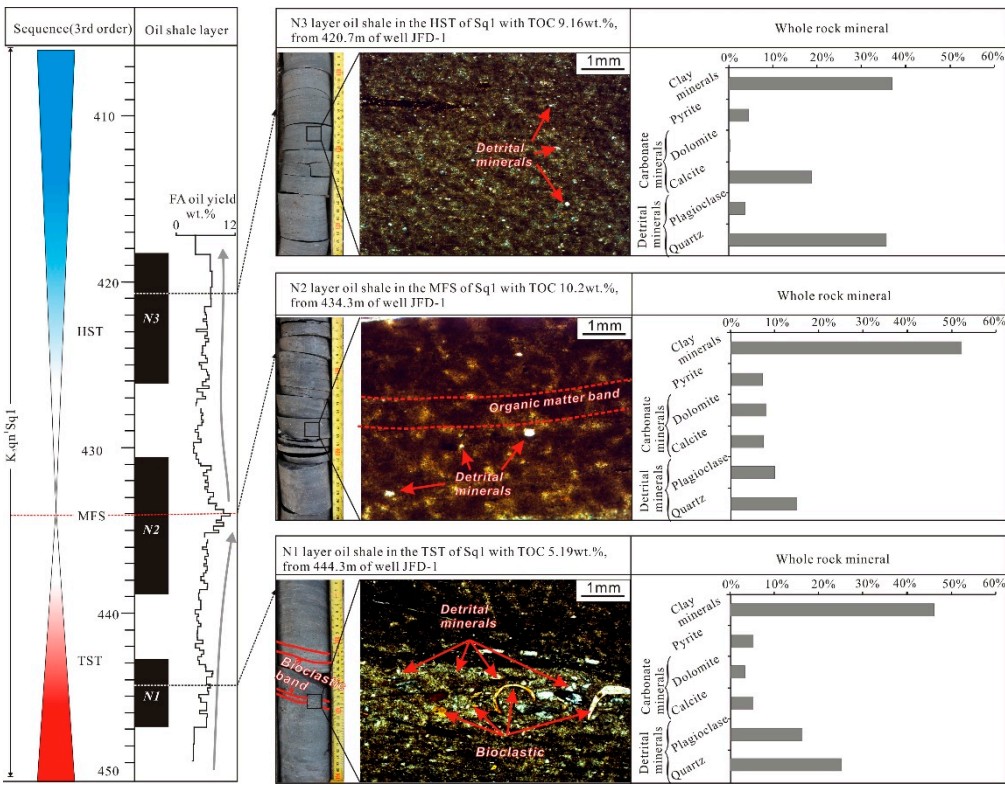

**Figure 9.** Diagram showing sequence stratigraphic divisions and variations in lithology, geochemistry (oil yield), and petrographic and mineralogical characteristics of oil shales of the $K_2qn^1$ member in the Songliao Basin.

## 5. Discussion

### 5.1. Origin of Organic Matter and Bioproductivity

The warm and humid paleoclimate during the period of oil shale deposition within the $K_2qn^1$ member yielded elevated levels of mineral nutrients that caused increased amounts of phytoplankton growth and improved biological productivity [27]. The N2 and N3 layers contain type-I kerogen that is dominated by lamalginite and telalginite, whereas the N1 oil shale layer contains type-I and -II kerogen that is dominated by alginate and sporinite. All of the oil shale samples analyzed during this study are immature to low maturity and have HI values that positively correlate with a transformation of the OM present in these samples from type-II to -I kerogens (i.e., from terrigenousto aquatic OM). The TOC and HI values for the N1 layer positively correlate ($r^2$ = 0.4; Figure 10a), whereas the N2 and N3 layers have TOC and HI data that do not correlate (Figure 10b,c). The data suggest that the OM enrichments within the N2 and N3 oil shale layers were not controlled by the origin of the OM within these units, contrasting with the data for the N1 oil shales that provide evidence for this relationship.

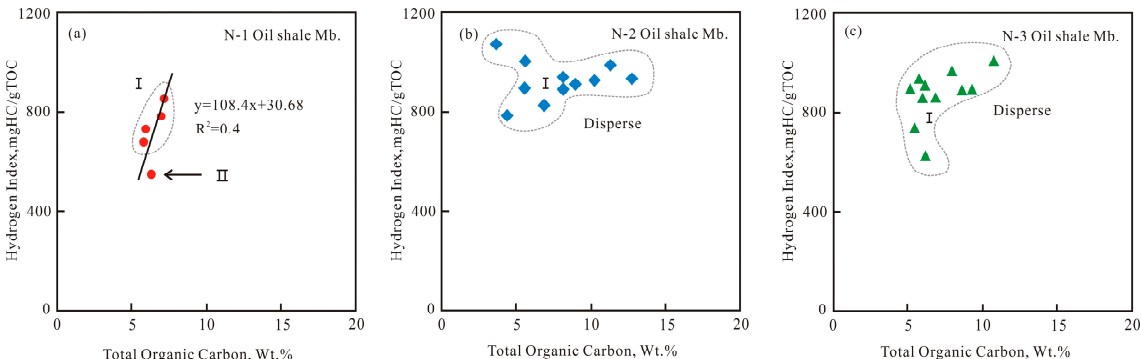

**Figure 10.** Plot of TOC versus Hydrogen index (HI) values for the (**a**) N1, (**b**) N2, and (**c**) N3 oil shale layers of the $K_2qn^1$ member of the Songliao Basin.

### 5.2. Effects of Mineral Dilution on Organic Matter Abundances

The amount of OM within an oil shale can be affected by the dilution effect of an increased abundance of terrestrial detritus. The majority of the quartz within the oil shale samples from the three wells analyzed during this study is detrital. Thus, we can use a detritus index (DI; calculated by summing the abundances of quartz, clay minerals, and feldspar) [28–32] to indicate variations in detrital influx. Generally, high DI values are indicative of times of high input of terrigenous material from continental sources and/or decreased carbonate productivity or an increase in the dissolution of carbonates [25,26].

The N2 and N3 oil shale layers have relatively low DI values relative to the N1 oil shale layers and the mudstones in the study area. This suggests that the periods of N2 and N3 deposition were characterized by low clastic sedimentary input to the lake. Furthermore, the low DI was probably a result of the presence of dense vegetation that acted as traps or baffles in the drainage area associated with the lake or an increase in the water level within the lake [33,34]. The negative correlation between DI and TOC values ($R^2 = -0.53$; Figure 11a) within these units is associated with a positive correlation between TOC values and carbonate mineral abundances ($R^2 = 0.43$; Figure 11b), indicating that terrigenous clastic material can clearly dilute the amount of OM within these sediments. The deposition of the thick and high-quality N2 and N3 oil shale layers that contain elevated carbonate mineral abundances was influenced by variations in lake level and climate change, with both layers being deposited during high lake level Sq1 conditions. In comparison, the thin, lower quality N1 oil shale layer that contains abundant detrital minerals was deposited during a TST. In addition, alkaline water is thought to favor the presence of abundant $CO_3^{2-}$ ions [35,36], suggesting that the N2 and N3 oil shale layers were deposited in an environment associated with a high lake level and alkaline waters, which is consistent with the TS values of these units. The lake-level fluctuation is related to the global sea level rising stage and periodic ingressions of ocean water as an alternative reason for lake-level high stands during early $K_2qn^1$. The increase of salinity caused by ingressions of ocean water may also be the reason for the increase of carbonate content during the high lake level [15,22]. Moreover, the high content of carbonate minerals is probably caused by algal blooms inducing carbonate precipitation [2].

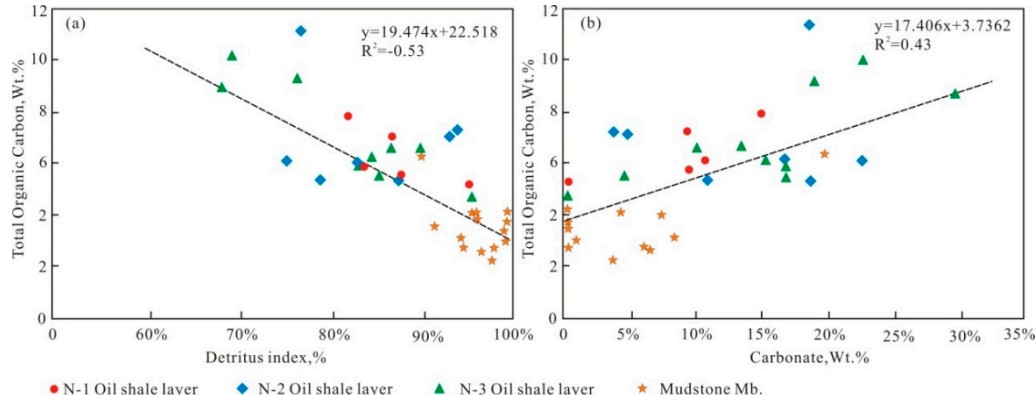

**Figure 11.** Plots of (**a**) TOC versus detritus index and (**b**) TOC versus carbonate content for the $K_2qn^1$ member of the Songliao Basin.

### 5.3. Thermal Evolution of Organic Matter

The thermal evolution of the source rocks in the study area has been analyzed using data such as vitrinite reflectance (Ro) and rock pyrolysis values. The oil shales in the study area that were buried to a depth of 440–460 m (Ro values of 0.48–0.50%; $T_{max}$ values of 435–446 °C) are immature to low maturity [25,26]. In comparison, oil shales buried to depths of 700–1080 m (Ro values of 0.51–0.55%; $T_{max}$ values of 435–446 °C) are low maturity and are slightly below the conditions of the oil generation window. The HCI and $S_1/(S_1 + S_2)$ values for these oil shales increase with increasing depth to ca. 080 m, indicating a slight increase in the free hydrocarbon content of these rocks (Figure 12).

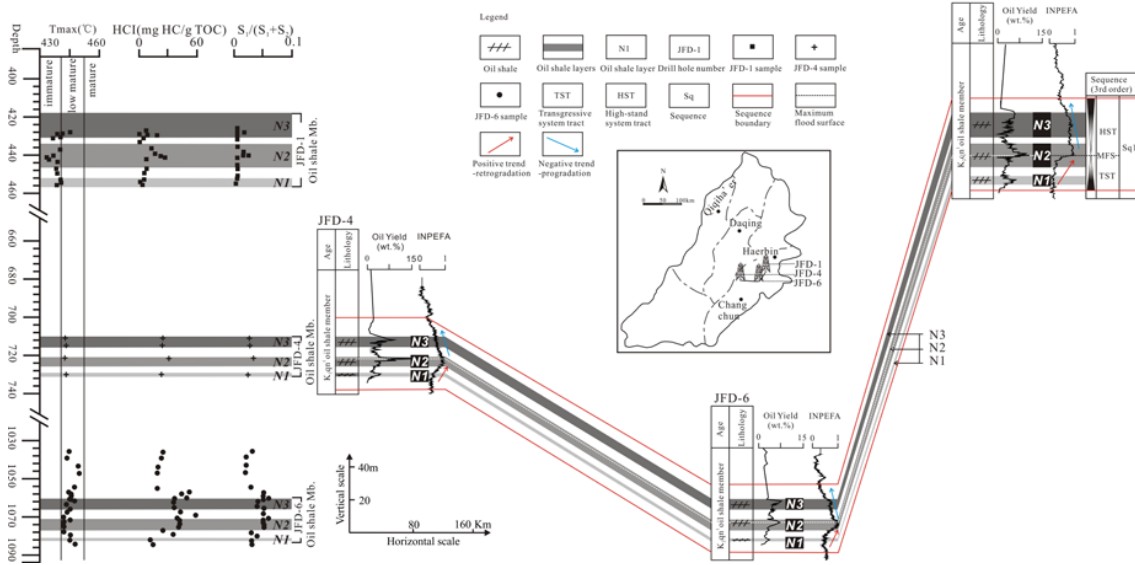

**Figure 12.** Diagram showing sequence stratigraphic correlations between oil shale layers and variations in the thermal evolution of the $K_2qn^1$ member of the Songliao Basin.

The thermal evolution of oil shales is a key factor in the hydrocarbon generation of these rocks during ICP. Hydrocarbon generation modeling using type-I and -II organic matter within non-marine shales indicates that vitrinite reflectance (Ro) values of 0.5% are associated with OM that has 90% of the original hydrocarbon generation potential. However, oil shale with Ro values of >1.0% are associated with a rapid decline in hydrocarbon generation potential. This indicates that oil shales buried to depths of <1000 m within the Songliao Basin have significant ICP potential. In addition, the characteristics of medium to low maturity organic shales in China and the technology used in earlier oil shale resource

developments has enabled the efficient application of underground horizontal well electric heating upgrading type technology [1]. This technology is suitable for oil shale-rich areas with burial depths of 300–3000 m and is expected to greatly expand the potential development and utilization of deep oil shale resources (depths > 1000 m).

### 5.4. Hydrocarbon Generation Potential

Plots of the petroleum potential (PP)−TOC, HI−TOC, and $S_2$−TOC can reveal the potential for oil shale to produce hydrocarbons [36]. Oil shale samples of the $K_2qn^1$ have PP values from 33 to 122 mgHC/g, and HI values of 687–1080 mg/g. The data are shown in PP−TOC and HI−TOC diagrams in Figure 13a,b and Table 1. The three oil shale layers deposited in $K_2qn^1$ is a good source rock, indicating that the oil shale deposited during this stage is prone to generate oil during the ICP.

This can be further quantified using the source potential index (SPI) [37].

$$SPI = (S_1 + S_2) \times h \times p/1000 \tag{1}$$

where $h$ (m) is thickness and $p$ (t/m$^3$) is bulk density. SPI values were calculated in order to quantify the amount of potential hydrocarbons within the study area. Using a density [38] of 2.22 t/m$^3$, the N3 (SPI = 0.88 tHC/m$^2$) and N2 (SPI = 0.92 tHC/m$^2$) oil shale layers have high SPV values, whereas the N1 oil shale layer has a low SPI value (0.21 tHC/m$^2$), indicating that the N2 and N3 layers should be considered highly prospective development targets.

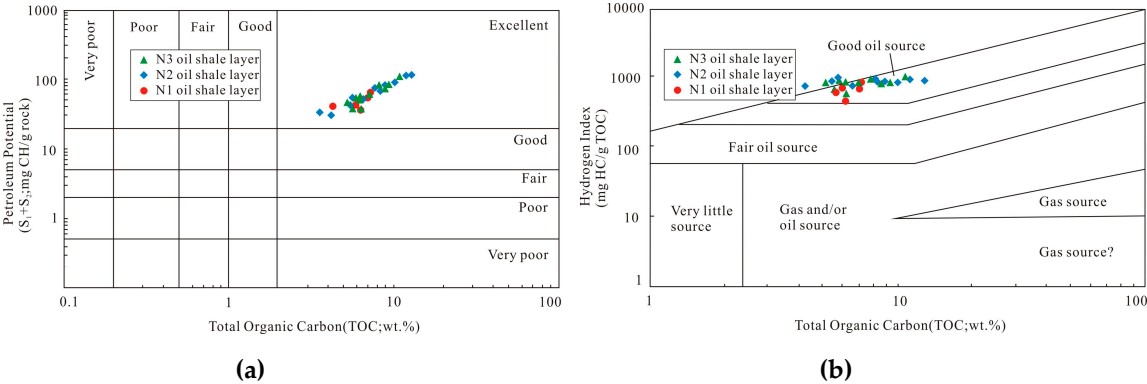

**Figure 13.** Plots of (**a**) HI vs. T$_{max}$ [36] and (**b**) TOC vs. S$_2$ [36] showing the kerogen typesfor oil shales of the $K_2qn^1$ member of the Songliao Basin.

**Table 1.** Summary of the characteristics of the oil shale samples analyzed during Rock–Eval pyrolysis, total sulfur, total organic carbon, and vitrinite reflectance values.

| Depth(m) | Sample | Lithology | Well | Oil Shale Layer | S1 | S2 | Tmax | $S_1/(S_1 + S_2)$ | HCI | HI | TS | TOC | VR |
|---|---|---|---|---|---|---|---|---|---|---|---|---|---|
| | | | | | mg/g | mg/g | °C | | mg/g TOC | mg/g TOC | wt.% | wt.% | %Ro |
| 420.70 | B1-97 | oil shale | JFD-1 | N3 | 1.86 | 80.03 | 446.00 | 0.02 | 20.31 | 873.69 | 0.93 | 9.16 | |
| 421.23 | B1-99 | oil shale | JFD-1 | N3 | 0.52 | 39.41 | 442.00 | 0.01 | 9.51 | 720.48 | 0.62 | 5.47 | 0.48 |
| 423.34 | B1-95 | oil shale | JFD-1 | N3 | 0.41 | 37.95 | 442.00 | 0.01 | 6.62 | 613.09 | 0.57 | 6.19 | |
| 429.40 | B1-98 | oil shale | JFD-1 | N3 | 0.53 | 37.81 | 442.00 | 0.01 | 15.14 | 1080.29 | 0.52 | 3.50 | 0.5 |
| 432.80 | B1-91 | oil shale | JFD-1 | N2 | 1.24 | 54.67 | 438.00 | 0.02 | 18.70 | 824.59 | 0.76 | 6.63 | |
| 433.93 | JFD1-B1 | oil shale | JFD-1 | N2 | 1.45 | 56.89 | 435.00 | 0.02 | 25.80 | 1012.28 | 0.61 | 5.62 | 0.48 |
| 434.33 | JFD1-B2 | oil shale | JFD-1 | N2 | 2.95 | 94.97 | 436.00 | 0.03 | 28.92 | 931.08 | 1.08 | 10.20 | |
| 434.53 | JFD1-139 | oil shale | JFD-1 | N2 | 1.96 | 74.21 | 438.00 | 0.03 | 24.02 | 909.44 | 0.46 | 8.16 | |
| 435.70 | B1-88 | oil shale | JFD-1 | N2 | 0.44 | 33.13 | 441.00 | 0.01 | 10.40 | 783.22 | 0.34 | 4.23 | |
| 442.63 | JFD1-186 | oil shale | JFD-1 | N1 | 0.65 | 55.55 | 444.00 | 0.01 | 9.22 | 787.94 | 0.67 | 7.05 | |
| 445.70 | B1-81 | oil shale | JFD-1 | N1 | 0.44 | 34.40 | 442.00 | 0.01 | 7.09 | 553.95 | 0.57 | 6.21 | |
| 712.70 | JFD4-53 | oil shale | JFD-4 | N3 | 2.09 | 75.25 | 443.00 | 0.03 | 24.19 | 870.95 | 0.87 | 8.64 | |
| 714.1 | JFD4-60 | oil shale | JFD-4 | N3 | 1.57 | 58.00 | 443.00 | 0.03 | 22.85 | 844.25 | 0.57 | 6.87 | |
| 721.7 | JFD4-98 | oil shale | JFD-4 | N2 | 3.74 | 118.37 | 442.00 | 0.03 | 29.45 | 932.05 | 1.16 | 12.70 | 0.51 |
| 729.1 | JFD4-135 | oil shale | JFD-4 | N1 | 1.60 | 61.32 | 443.00 | 0.03 | 22.38 | 857.62 | 0.85 | 7.15 | |
| 1058.60 | JFD6-72 | oil shale | JFD-6 | N3 | 5.42 | 105.57 | 445.00 | 0.05 | 50.42 | 982.05 | 1.08 | 10.75 | |
| 1059.60 | JFD6-73 | oil shale | JFD-6 | N3 | 5.00 | 75.38 | 445.00 | 0.06 | 62.66 | 944.61 | 0.67 | 7.98 | |
| 1060.80 | JFD6-76 | oil shale | JFD-6 | N3 | 4.17 | 50.79 | 444.00 | 0.08 | 69.27 | 843.69 | 0.83 | 6.02 | |
| 1060.90 | JFD6-77 | oil shale | JFD-6 | N3 | 2.96 | 55.25 | 446.00 | 0.05 | 48.01 | 896.19 | 0.65 | 6.17 | 0.55 |
| 1061.00 | JFD6-B1 | oil shale | JFD-6 | N3 | 3.57 | 53.16 | 445.00 | 0.06 | 61.45 | 914.97 | 0.66 | 5.81 | |
| 1061.90 | JFD6-78 | oil shale | JFD-6 | N3 | 1.87 | 45.12 | 447.00 | 0.04 | 36.24 | 874.42 | 1.03 | 5.16 | |
| 1069.40 | JFD6-h85 | oil shale | JFD-6 | N2 | 4.50 | 75.56 | 443.00 | 0.06 | 55.76 | 936.31 | 1.20 | 8.07 | |
| 1069.90 | JFD6-h86 | oil shale | JFD-6 | N2 | 6.84 | 111.45 | 441.00 | 0.06 | 60.80 | 990.67 | 0.97 | 11.25 | |
| 1071.90 | JFD6-h88 | oil shale | JFD-6 | N2 | 2.11 | 48.00 | 445.00 | 0.04 | 39.29 | 893.85 | 0.61 | 5.37 | 0.53 |
| 1079.90 | JFD6-96 | oil shale | JFD-6 | N1 | 1.49 | 37.10 | 445.00 | 0.04 | 35.14 | 875.00 | 0.78 | 4.24 | |
| 1079.4 | JFD6-95 | oil shale | JFD-6 | N1 | 2.24 | 39.94 | 444.00 | 0.05 | 38.55 | 687.44 | 0.75 | 5.81 | |
| 435.70 | B1-88 | mudstone | JFD-1 | | 0.44 | 33.13 | 441.00 | 0.01 | 10.40 | 783.22 | 0.77 | 4.23 | |
| 441.49 | B1-85 | mudstone | JFD-1 | | 0.16 | 13.14 | 440.00 | 0.01 | 7.27 | 597.27 | 0.32 | 2.20 | |
| 446.50 | JFD1-18 | mudstone | JFD-1 | | 0.10 | 13.51 | 442.00 | 0.01 | 4.44 | 600.44 | 0.42 | 2.25 | |
| 1038.85 | JFD6-67 | mudstone | JFD-6 | | 0.88 | 34.38 | 443.00 | 0.02 | 23.50 | 918.02 | 0.64 | 3.75 | |
| 1054.70 | B6-6 | mudstone | JFD-6 | | 0.42 | 13.06 | 447.00 | 0.03 | 18.46 | 574.07 | 0.62 | 2.28 | |
| 1063.90 | JFD6-80 | mudstone | JFD-6 | | 1.35 | 32.34 | 443.00 | 0.04 | 34.57 | 828.17 | 0.73 | 3.91 | |
| 1065.90 | JFD6-82 | mudstone | JFD-6 | | 0.91 | 17.26 | 445.00 | 0.05 | 35.55 | 674.22 | 0.51 | 2.56 | |
| 1072.90 | JFD6-89 | mudstone | JFD-6 | | 1.40 | 27.20 | 441.00 | 0.05 | 40.76 | 791.85 | 0.57 | 3.44 | |
| 1074.90 | JFD6-91 | mudstone | JFD-6 | | 1.21 | 22.34 | 441.00 | 0.05 | 41.37 | 763.76 | 0.73 | 2.93 | |
| 1075.90 | JFD6-92 | mudstone | JFD-6 | | 1.42 | 26.10 | 441.00 | 0.05 | 40.23 | 739.38 | 0.68 | 3.53 | |

Abbreviations as follows: HCI = pyrolysis hydrocarbon index, HI = hydrogen index, TS = total sulfur, TOC = total organic carbon, and VR = vitrinite reflectance.

### 5.5. Rock Brittleness

Rock brittleness is a key control on fracture development during ICP [39], with increased abundances of brittle minerals associated with corresponding changes in rock mechanical parameters, which involves a decrease in Poisson's ratio and an increase in Young's modulus values [40]. Here, we classify quartz, feldspar, and carbonate as brittle minerals, with increasing quartz and feldspar abundances increasing brittleness and increasing carbonate abundances associated with improvements during later transformation. Reservoirs containing abundant quartz or carbonate are conducive to the generation of complex fracture networks, whereas plastic formations containing elevated amounts of clay minerals are not conducive to the formation of complex fracture networks. This indicates that elevated brittle mineral abundances are a key factor in obtaining high yields by fracturing. All three oil shale layers contain similar distribution pattern of quartz and feldspar content (peak with 20–25% and 10–15%, respectively; Figure 14a,b). However, these three oil shale layers contain obviously different amounts of carbonate, with the N2 and N3 layers mainly containing 15–20% carbonate, whereas the N1 layer mainly contains 5–10% carbonate (Figure 14c).

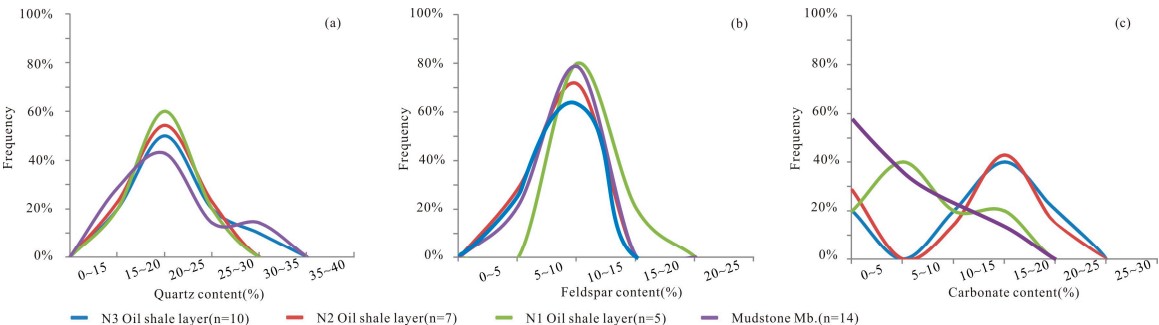

**Figure 14.** Frequency accumulation curves showing variations in (**a**) quartz, (**b**) feldspar, and (**c**) carbonate abundances in the $K_2qn^1$ member of the Songliao Basin.

Previous research has obtained brittleness index (BI) values using mineral abundances and the following formula:

$$BI = (QZ + Car + Fels)/(QZ + Car + Fels + Clay) \times 100\% \tag{2}$$

where *BI* is the brittleness index, *QZ* is the quartz content, *Car* is the carbonate content, *Fels* is the feldspar content, and *Clay* is the total clay content. The mudstone and N1 oil shale layers in the study area have BI values that are generally <30% and between 30% and 40%, respectively (Figure 15), indicating that these units are not suitable for fracturing. In comparison, the N2 and N3 oil shale layers have higher *BI* values (generally 40–50%), with the N2 layer in particular having elevated brittleness values. The sequence stratigraphy and geochemical data obtained during this study indicate that the N2 and N3 layers were deposited under high lake level conditions and are associated with alkaline waters and ingressions of ocean water, all of which enriched these layers in carbonates. However, the N1 layer was deposited in a TST environment associated with large amounts of terrigenous clastic input that significantly decreased carbonate productivity and dissolution. The data indicate that an increase in lake level and the development of alkaline waters were key factors in the formation of the high carbonate content and high brittleness index of the N2 and N3 oil shale layers.

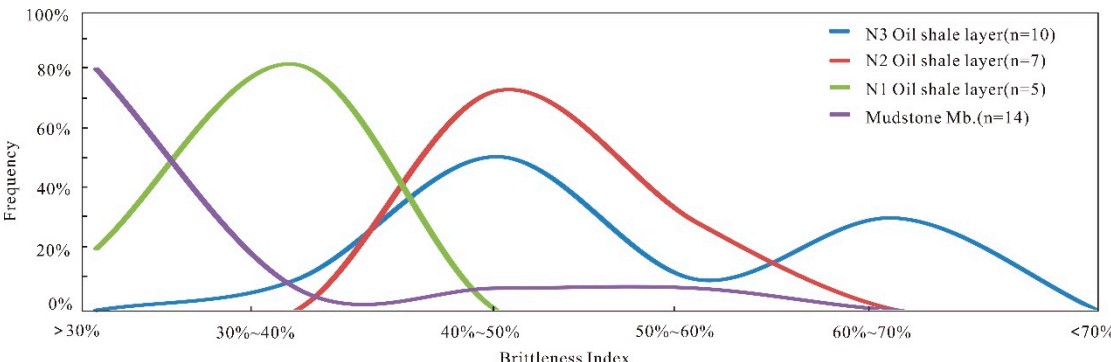

**Figure 15.** Frequency accumulation curve showing variations in brittleness index values in the $K_2qn^1$ member of the Songliao Basin.

*5.6. Geological Optimization of ICP*

Existing ICP technology for the exploitation of oil shales can be divided into four types by the types of heating used, which are electrical heating by conduction, radiation heating, gas thermal convection heating, and chemical reaction heating. The chemical reaction heating method has already been successful in the ICP of the $K_2qn^1$ member oil shales in the Songliao Basin [1,41]. This process involves the fracturing of high OM abundance oil shales to establish oil and gas channels before the injection of preheated mixed gases into the target horizon to cause thermochemical reactions within the oil shale. This causes the cracking of the oil shales, generating oil and gas [42]. All of this means that oil shales containing elevated amounts of OM and with high SPI and BI values are ideal targets for ICP.

A high-resolution sequence stratigraphic framework can sensitively reflect the change of eustatic lake level; furthermore, the variation of lake level greatly affects the organic matter abundance, organic matter type, and mineral composition of oil shales [1,4,5]. Therefore, the sequence stratigraphic framework established by spectral analysis can effectively predict the distribution of ICP-amenable layers. The sequence stratigraphic framework for the Qingshankou Formation based on spectral trend attribute analysis indicates that oil shales within this formation have a well-defined vertical distribution pattern. Thus, deposition of oil shale layers were controlled by the sequence stratigraphic framework of this area (Figure 16). Petrological and geochemical analysis indicates that the well-preserved dark oil shale layers deposited during high lake level conditions during Sq1 (i.e., layers N2 and N3) are ideal targets for ICP. These oil shale layers have higher BI values (generally 40–50%) and are thicker (maximum of 13.3 m; average of 12.0 m) than the N1 oil shale deposited during TST conditions in Sq1, indicating that the N2 and N3 layers are more susceptible to fracturing and likely to develop oil–gas channels than the N1 layer. The Sq1 N2 and N3 oil shale layers contain 5.2–23.9 wt % TOC, whereas the TST N1 oil shale layer has far lower TOC abundances(5.1–6.2 wt %). In addition, the N2 and N3 oil shale layers have higher SPI values (0.92 and 0.88 tHC/m$^2$, respectively) than the N1 layer (0.21 tHC/m$^2$). Finally, oil shales buried at depths of <1000 m have hydrocarbon generation capacities that are suitable for targeting using ICP. Sequence correlation for the oil shale layers within the Southeastern Uplift region of the Songliao Basin, indicating that the ICP-amenable N2 and N3 layers are widespread and have stable distribution throughout this region (Figure 12).

China's in situ conversion resources of organic-rich shales are mainly distributed in the non-marine shale beds in the Ordos Basin, the Songliao Basin, and the Junggar Basin. According to preliminary test data, the technically recoverable oil resources using underground in situ conversion are approximately $(700 - 900) \times 10^8$ t, the economically recoverable resources are about $(200 - 250) \times 10^8$ t when the oil price is USD $(377 - 409)/m^3$ (USD $(60 - 65)$/bbl) [1], which is equivalent to the total technically recoverable resources of conventional oil. Among them, the in situ conversion of oil shale with high organic matter content and stable distribution in the $K_2qn^1$ of the Songliso Basin is considerable realistic.

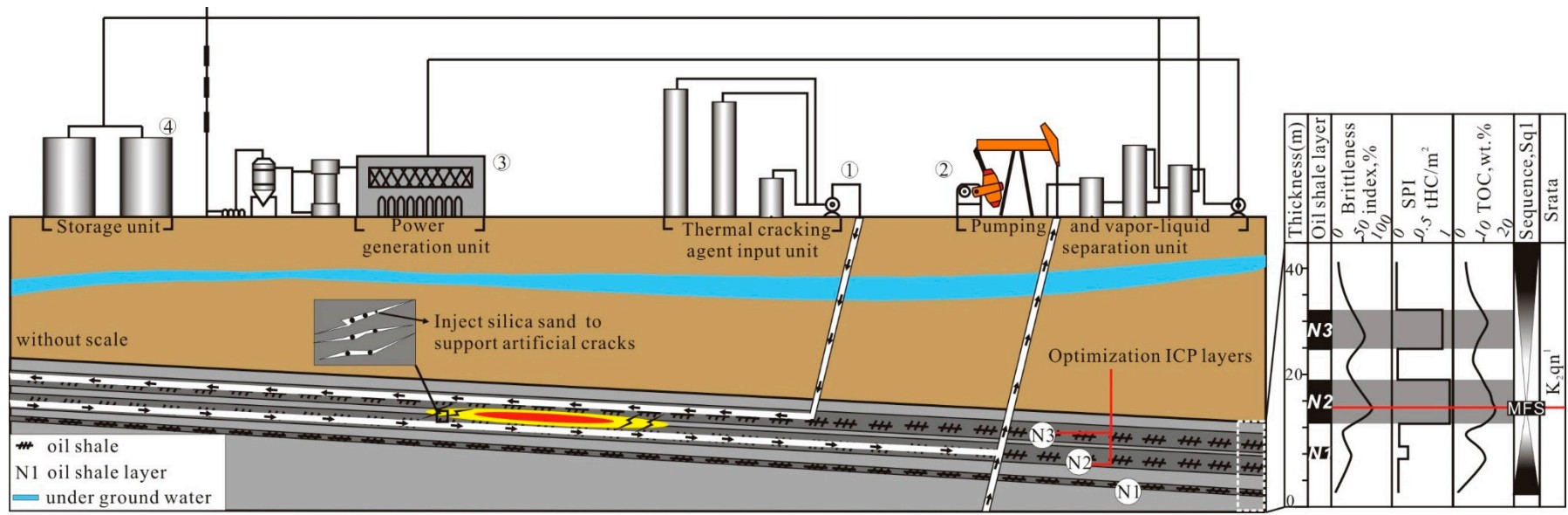

**Figure 16.** Model showing the possible in situ conversion processing (ICP) extraction of oil from the $K_2qn^1$ member oil shales of the Songliao Basin; adapted from [42].

## 6. Conclusions

The Songliao Basin contains three thick $K_2qn^1$ oil shale layers: (1) an N1 oil shale layer with a lower average thickness (2.1 m) and quality (average TOC of 6.3 wt %), (2) N2 (average oil shale thickness of 6.0 m, average TOC of 7.8 wt %), (3) N3 (average oil shale thickness of 6.2 m, average TOC of 6.7 wt %) oil shale layers within this member.

According to sequence stratigraphic correlations, these three main thick oil shale layers of the Sequence1 (Sq1) unit in $K_2qn^1$ are confirmed consistently present throughout the Southeastern Uplift region of the basin. The spectral trend attributes reflect that the lake reached a maximum flood surface of the $K_2qn^1$ in N2 oil shale layer, and it also indicated that the total organic carbon (TOC), and Fischer assay (FA) oil yield are significantly increasing. Oil shale layers (N2, N3) deposited during high lake level conditions associated with alkaline waters and ingressions of ocean water have high TOC (maximum of 23.9 wt %; average of 7.2 wt %) and aquatic OM abundances (maximum HI of 1080.2 mg/g; average of 889.9 mg/g) and elevated carbonate contents (maximum of 29.5%; average of 15.4%).

Petrological and geochemical analysis indicates that the well-preserved, dark, high lake level, Sq1 N2 and N3 oil shale layers represent highly prospective targets for ICP. These oil shale layers have higher BI values (generally 40–50%) and are thicker (maximum of 13.3 m; average of 12.0 m) than the Sq1 TST N1 oil shale layer, meaning that the N2 and N3 layers are ideally suited for fracturing and the development of oil–gas channels. The N2 and N3 oil shale layers also have much higher SPI values (0.92 and 0.88 tHC/m$^2$, respectively) than the N1 layer (0.21 tHC/m$^2$). The hydrocarbon-generating capacity of the oil shales also indicates that oil shale layers buried to depths of <1000 m are suitable for ICP.

**Author Contributions:** P.Z., Y.X. and Q.M. contributed to edit in this work, J.Z. and Y.X. provided the funding, L.S. and S.Z. made the formal analysis and samples resources, Q.M. and Z.L. made the validation. All authors have read and agreed to the published version of the manuscript.

**Funding:** We acknowledge support from the Opening Foundation of the Key Laboratory of Unconventional Petroleum geology, China Geological Survey. This study was financially supported by the China Geological Survey (Grant DD2019139-YQ19JJ04, Grant DD20189606), Program of the National Natural Science Foundation of China (Grant 41872103), and Evaluation of In Situ Development Resources and Dynamic Geological Parameters of Oil Shale in Jilin Province project (Grant 20180201077SF). Zhang Penglin also thanks the National Construction of High-quality University Projects of Graduates from China Scholarship Council (CSC)(No.201906170223) for an 12-months' scholarship.

**Conflicts of Interest:** The authors declare no conflict of interest.

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
