# Peer review of "Sequence Stratigraphy and Geochemistry of Oil Shale Deposits in the Upper Cretaceous Qingshankou Formation of the Songliao Basin, NE China: Implications for the Geological Optimization of In Situ Oil Shale Conversion Processing"

_energies, doi:10.3390/en13112964_

Round 1

Reviewer 1 Report

Comments for the Authors

Article ENERGIES-824468

“Sequence stratigraphy and geochemistry of oil shale deposits in the Upper Cretaceous Qingshankou Formation of the Songliao Basin, NE China: implications for the geological optimization of insitu oil shale conversion processing”

Article ENERGIES-824468, “Sequence stratigraphy and geochemistry of oil shale …” is interesting.  This has a lot of geologic and chemical analysis that I have little experience in assessing.  However, I believe the geological and chemical information should be good to know for other researchers. 

Line 54:  I’m not sure what this means, “…insitu conversion processing (ICP), which is an approach where oil is extracted insitu conversion processing from oil shales.”  Does it mean that “oil is converted from oil shales insitu?”  Or does it mean that “the oil separation process from oil shales is done insitu?”  They both end up meaning the same thing, but the way it reads is confusing. 

Normally insitu processes of the past were when a shale type rock that was hard would be heated up or burned insitu, and with very limited oxygen, in order to melt out the liquid oils and then pump them to the surface.  If you can explain this process better, that would be helpful.  It would be good just to know if you are using a heat process, or not using a heat process, along with chemicals, if the process is not confidential. 

Okay, Line 424 does start to explain the heating process.  Good. 

And Line 447 with the figure.  This is very good to show.  Any information on the amount of energy used to energy extracted would be good if you can include it.  This is often called Energy Return on (energy) Investment (EROI).  However the figure gives a good indication of the processes involved.   

Line 91 says, “shales richin organic matter,”  it should be “shales rich in organic matter,”  Also Line 169: “includethe” and “fromthe” need to be separated.

Author Response

Answer to the question asked by the first reviewer:

Dear reviewer:

We have revised the manuscript according to the suggestions. Thank you for your review of this article and your valuable comments. The following are the answers or explanations to the questions asked by you.

Question1: Line 54:  I’m not sure what this means, “…insitu conversion processing (ICP), which is an approach where oil is extracted insitu conversion processing from oil shales.”  Does it mean that “oil is converted from oil shales insitu?”  Or does it mean that “the oil separation process from oil shales is done insitu?”  They both end up meaning the same thing, but the way it reads is confusing. 

Normally insitu processes of the past were when a shale type rock that was hard would be heated up or burned insitu, and with very limited oxygen, in order to melt out the liquid oils and then pump them to the surface.  If you can explain this process better, that would be helpful.  It would be good just to know if you are using a heat process, or not using a heat process, along with chemicals, if the process is not confidential.

Answer---- by consulting related patents and articles, we have a more explicit explanation of ICP technology. Since some technologies are still in a state of confidentiality, it is currently not possible to provide chemical reagents related to oil shale heating involved in ICP. However, it is certain that heating is the key factor in the ICP process. So we explained the ICP in introduction as follows:

In-situ conversion processing (ICP) of oil shale underground at the depth ranging from 300 m to 3 000 m is a physical and chemical process caused by using horizontal drilling and electric heating technology, which converts heavy oil, bitumen and various organic matter into light oil and gas in a large scale, which can be called “underground refinery”.

Question2: And Line 447 with the figure.  This is very good to show.  Any information on the amount of energy used to energy extracted would be good if you can include it.  This is often called Energy Return on (energy) Investment (EROI).  However the figure gives a good indication of the processes involved.

Answer---- Energy Return on (energy) Investment (EROI) is indeed a significant parameter for reaction energy conversion, but we are sorry that we cannot provide this parameter at this stage. Information on energy inputs is currently not available through official channels from oil companies. As the research progresses in further, we will provide more information in this regard in the future, Thank you for this meaningful suggestion.

In order to discuss the practical economic and production value by this energy conversion process, the relationship between crude oil price and the amount of recoverable resources is discussed.

The specific changes in Section 5.6 are as follows: China’s in-situ conversion resources of shale oil are mainly distributed in the nonmarine shale beds in the Ordos Basin, the Songliao Basin, and the Junggar Basin. According to preliminary test data, the technically recoverable oil resources using underground in-situ conversion are approximately (700-900)×108 t, the economically recoverable resources are about (200-250)×108 t when the oil price is USD (377-409)/m3 (USD (60-65)/bbl), which is equivalent to the total technically recoverable resources of conventional oil. Among them, the in-situ conversion of oil shale with high organic matter content and stable distribution in the K2qn1 of the Songliso Basin is considerable realistic.

Question3: Line 91 says, “shales richin organic matter,” it should be “shales rich in organic matter,” Also.

Answer----we have revised the words.

Question4: Line 169: “includethe” and “fromthe” need to be separated.

Answer---- we have revised the words. It is true that many words have similar errors due to WORD version problems. We have corrected them

Thank you again for your suggestions, which were all very helpful for improving our paper. We also sincerely appreciate Energies and you for giving us the opportunity to revise our manuscript.

Reviewer 2 Report

It is judged that this study has a theoretical and practical significance in that it is based on empirical data. However, it is required to revise the following points.

First, the summary is too long. It is necessary to abbreviate around the core content.

Second, a theoretical discussion related to this study is needed.

Third, it is necessary to present the limitations of the methodology adopted in this study.

Author Response

Answer to the question asked by the second reviewer:

Dear reviewer:

We have revised the manuscript according to the suggestions. Thank you for your review of this article and your valuable comments. The following are the answers or explanations to the questions asked by you.

Question1: Line 54:  I’m not sure what this means, “…insitu conversion processing (ICP), which is an approach where oil is extracted insitu conversion processing from oil shales.”  Does it mean that “oil is converted from oil shales insitu?”  Or does it mean that “the oil separation process from oil shales is done insitu?”  They both end up meaning the same thing, but the way it reads is confusing. 

Normally insitu processes of the past were when a shale type rock that was hard would be heated up or burned insitu, and with very limited oxygen, in order to melt out the liquid oils and then pump them to the surface.  If you can explain this process better, that would be helpful.  It would be good just to know if you are using a heat process, or not using a heat process, along with chemicals, if the process is not confidential.

Answer---- by consulting related patents and articles, we have a more explicit explanation of ICP technology. Since some technologies are still in a state of confidentiality, it is currently not possible to provide chemical reagents related to oil shale heating involved in ICP. However, it is certain that heating is the key factor in the ICP process. So we explained the ICP in introduction as follows:

In-situ conversion processing (ICP) of oil shale underground at the depth ranging from 300 m to 3 000 m is a physical and chemical process caused by using horizontal drilling and electric heating technology, which converts heavy oil, bitumen and various organic matter into light oil and gas in a large scale, which can be called “underground refinery”.

Question2: And Line 447 with the figure.  This is very good to show.  Any information on the amount of energy used to energy extracted would be good if you can include it.  This is often called Energy Return on (energy) Investment (EROI).  However the figure gives a good indication of the processes involved.

Answer---- Energy Return on (energy) Investment (EROI) is indeed a significant parameter for reaction energy conversion, but we are sorry that we cannot provide this parameter at this stage. Information on energy inputs is currently not available through official channels from oil companies. As the research progresses in further, we will provide more information in this regard in the future, Thank you for this meaningful suggestion.

In order to discuss the practical economic and production value by this energy conversion process, the relationship between crude oil price and the amount of recoverable resources is discussed.

The specific changes in Section 5.6 are as follows: China’s in-situ conversion resources of shale oil are mainly distributed in the nonmarine shale beds in the Ordos Basin, the Songliao Basin, and the Junggar Basin. According to preliminary test data, the technically recoverable oil resources using underground in-situ conversion are approximately (700-900)×108 t, the economically recoverable resources are about (200-250)×108 t when the oil price is USD (377-409)/m3 (USD (60-65)/bbl), which is equivalent to the total technically recoverable resources of conventional oil. Among them, the in-situ conversion of oil shale with high organic matter content and stable distribution in the K2qn1 of the Songliso Basin is considerable realistic.

Question3: Line 91 says, “shales richin organic matter,” it should be “shales rich in organic matter,” Also.

Answer----we have revised the words.

Question4: Line 169: “includethe” and “fromthe” need to be separated.

Answer---- we have revised the words. It is true that many words have similar errors due to WORD version problems. We have corrected them

Thank you again for your suggestions, which were all very helpful for improving our paper. We also sincerely appreciate Energies and you for giving us the opportunity to revise our manuscript.

Reviewer 3 Report

The article contains extensive geochemical research relevant to shale oil resources. World reserves of unconventional oil deposits are not precisely defined. All information on geological resources should be treated as estimates. Preliminary analyzes of the determination of the size of the resources show that unconventional oil deposits far exceed those of conventional deposits. Exploitation of this type of deposits may meet the demand for liquid fuels in the future. In the world oil industry, the topic of unconventional deposits, in particular shale oil, is gaining more and more importance. At present oil production from shale oil deposits is negligible compared to the total number of resources. ICP analysis is well done. Below are a few points that need to be improved.

Citations in the text should be written in square brackets and not in superscripts.
Section 3.1. The title should be extended.
Section 3.3. Change the title, but most of all delete or replace the word "Whole".
Figure 2 should display: a, b, c. These signatures should be added in the drawings and in the description.
Titles 4.1. and 4.2 should be extended.
In point 4.2. describe the apparatus.
In Lines: 213 and 303 - the sentence begins with points – it should be improve.
In section 4.4. the sentence 'All of these samples have relatively high total sulfur (TS)' should be reworded.
In lines: 84, 104, 155, 181, 190, 196, 198, 247, 256, 258, 259, 351, 352, 454, 456 - use a space between letters and units.
Figures 9 and 16 should be enlarged.
Figure 10 - the description of the points is missing: a, b, c.
Figure 13 - the description of the points is missing: a, b.
Drawings: 14 and 15 - unify units on the axes.
On line 406, add the number of formula.
In point 5.6, the economic benefits resulting from the conducted research should be added.
The literature has an incorrect layout.

Author Response

Answer to the question asked by the third reviewer:

Dear reviewer:

We have revised the manuscript according to the suggestions. Thank you for your detailed review of this article and your valuable comments. The following are the answers or explanations to the questions asked by you.

Question1: Citations in the text should be written in square brackets and not in superscripts.

Answer---- We have modified the reference format as required.

Question2: Section 3.1. The title should be extended.

Answer---- We have revised the title of Section 3.1., as Materials and sampling location.

Question3: Section 3.3. Change the title, but most of all delete or replace the word "Whole".

Answer---- We have revised the title of Section 3.3., as X-ray diffraction analysis.

Question4: Figure 2 should display: a, b, c. These signatures should be added in the drawings and in the description.

Answer---- We have revised the Figure 2, and add the signatures in the description of Section 3.1 and Section 4.1.

Question5: Titles 4.1. and 4.2. should be extended.

Answer---- We have revised the title of Section 4.1., as Lithostratigraphy of K2qn1. We have revised the title of Section 4.2., as Whole-rock powder analysis

Question6: In point 4.2. describe the apparatus.

Answer---- We have revised the point, and a description of the apparatus is added at the beginning of this section

Question7: In Lines: 213 and 303 - the sentence begins with points – it should be improve.

Answer---- We have deleted the point at the beginning of the sentence.

Question8: In section 4.4. the sentence 'All of these samples have relatively high total sulfur (TS)' should be reworded.

Answer---- In order to express the meaning more clearly and concisely, we have revised the sentence as “A relatively high content of total sulfur (TS) occurs in K2qn1, arguing for relatively high alkalinity.”.

Question9: In lines: 84, 104, 155, 181, 190, 196, 198, 247, 256, 258, 259, 351, 352, 454, 456 - use a space between letters and units.

Answer----We have revised the sentences with a space between letters and units.

Question10: Figures 9 and 16 should be enlarged..

Answer----We have revised the Figures 9 and 16. In addition, figure 16 shows the horizontal version

Question11: Figure 10 - the description of the points is missing: a, b, c.

Answer----We have added the description of the points (a,b,c) of Figures 10.

Question12: Figure 13 - the description of the points is missing: a, b.

Answer----We have added the description of the points (a,b) of Figures 13.

Question13: Drawings: 14 and 15 - unify units on the axes.

Answer----We have revised the Figures 14 and 15. In addition, a scale is added to the horizontal axis in FIG. 15.

Question14: On line 406, add the number of formula.

Answer----We have add the number of formula.

Question15: In point 5.6, the economic benefits resulting from the conducted research should be added.

Answer---- In point 5.6, we discussed the total amount of crude oil that China could obtain through ICP technology, and the relationship between crude oil price and the amount of recoverable resources is also discussed. As the price of crude oil continues to change recently, we hope to demonstrate the great potential of ICP technology through such discussions.

The specific revision is as follows: China’s in-situ conversion resources of organic-rich shale are mainly distributed in the nonmarine shale beds in the Ordos Basin, the Songliao Basin, and the Junggar Basin. According to preliminary test data, the technically recoverable oil resources using underground in-situ conversion are approximately (700-900)×108 t, the economically recoverable resources are about (200-250)×108 t when the oil price is USD (377-409)/m3 (USD (60-65)/bbl), which is equivalent to the total technically recoverable resources of conventional oil. Among them, the in-situ conversion of oil shale with high organic matter content and stable distribution in the K2qn1 of the Songliso Basin is considerable realistic.

Question16: The literature has an incorrect layout.

Answer---- The layout of references in this paper has been modified as required.

Thank you again for your suggestions, which were all very helpful for improving our paper. We also sincerely appreciate Energies and you for giving us the opportunity to revise our manuscript.

Reviewer 4 Report

Dear Authors,

The topic of the article is interesting and intelligible. The manuscript has certainly potential to improve. In my humble opinion, if the manuscript is thoroughly revised and reorganized, it can make a fine publication. To help improve the quality of this manuscript, I have added more comments bellow:

General Comments:

  1. Incorrect citation throughout the text. It is correct e.g. “of oil generated during the middle part of this test test [1]”…
  2. Check spaces throughout all paper (a lot of connected words in the text ...).
  3. Write the individual contribution of each author in "Author Contributions".
  4. The font size of the text in the title of the Figures throughout the paper should be adjusted in accordance with the "Instructions for Authors" (e.g. L-96; L-192…).
  5. Insufficient cited literature of technology ICP in the "Introduction". Please cite relevant literature.
  6. Correct the references in the "References" in accordance with the "Instructions for Authors".
  7. Too long "Abstract" needs to be shortened.

L25-29 “The N2 and N3oil shale layers within the Sq1 unit were deposited in a high lake level environment associated with ingressions of ocean water, generating oil shales with high TOC(maximum of 23.9 wt.%;average of 7.2 wt.%),abundance of aquatic organic matter (OM)(maximum Hydrogen index(HI)of 1080.2mg/g; average of 889.9mg/g)  and carbonate contents(maximum of 29.5 %;average of 15.4 %).” -> sentence too long, please rephrase into two sentences.

L29-33 “The N2 and N3 oil shale layers have…” .” -> sentence too long, please rephrase into two sentences.

L37-38 Keywords:… -> without bold

L43 “Chinese oil shale resources are” -> Chinese oil shale reserves are…

L44 “billion tons of shale that can yield 61 billion tons of shale oil,” -> billion tons of oil 61 billion tons of shale oil, of which recoverable reserves are 61 billion tons of oil.

L44 “, with the Upper Cretaceous of the…” -> Upper Cretaceous of the…

L46 “north eastern China crops out over an area of ca 26x104 km2 and is covered by 559x104 hectares of…” -> north eastern China crops out over an area of 260 000 km2 and is covered over 5 590 000 hectares of…

L47-50 “The basin records a rapid and large-scale lake transgression event during the deposition of…” -> sentence too long, please rephrase into two sentences.

L55 “Research into the insitu conversion…” -> Research into ICP…

L60 “University in cooperation with Israel”-> company name from Israel?

L60 “, leading to a pilot test of the in situ conversion processing of oil” -> , leading to a pilot test of ICP of oil…

L69-72 “However, since K 2 qn 1 are dominated by dark…” -> sentence too long, please rephrase into two or more sentences.

L112-115 “. This approach is based on a cyclical stratigraphic theoretical basis and converts well…” -> sentence too long, please rephrase into two or more sentences.

L118-120 “ However, significant changes in eustatic lake level during the deposition of the…” -> sentence too long, please rephrase into two sentences.

L120 define "GR"

L120-124 “In…” -> sentence too long, please rephrase into two sentences.

L130-132 “stratigraphic continuity where negative peaks ( i.e., large negative error s ) represent possible flooding surfaces , positive peaks (large positive error s ) represent possible sequence boundaries , and different peaks (different errors) are…” -> stratigraphic continuity: negative peaks represent possible flooding surfaces , positive peaks represent possible sequence boundaries, and different peaks  are…

L213 delete the point at the beginning

L234 delete point at the end of the subtitle

L246-249 “The average ‘‘true’’ HI values for the N3…” -> sentence too long, please rephrase into two or more sentences.

L261 delete point at the end of the subtitle

L265-269 “Comparing the GR curves…” -> sentence too long, please rephrase into two or more sentences.

L278-282 “in the form of 1) a rapid decrease in GR curve 278 data , 2) low wavelength but high amplitude MESA data where amplitude peaks occur at 100 ka intervals, indicating that these short term cycles are predominantly controlled by eccentricity variationsvariations11 3) negative peaks in PEFA curve data, and 4) INPEFA curve values close to one that represent a negative turning point from positive to negative trends” -> Rearrange text: in the form:

1) a rapid decrease in GR curve 278 data,

2) low wavelength but high amplitude MESA data where amplitude peaks occur at 100 ka intervals, indicating that these short term cycles are predominantly controlled by eccentricity variationsvariations11

3) negative peaks in PEFA curve data,

4) INPEFA curve values close to one that represent a negative turning point from positive to negative trends.

L282-285 “The TST are dominated by dark…” -> sentence too long, please rephrase into two sentences.

L315 In Figure 10a, add the corresponding equation y =kx+l.

L319-324 “The majority of the quartz within the oil shale samples from the three…” -> sentence too long, please rephrase into two or more sentences.

L326-329 “This suggests that the periods of N2 and N3 deposition were…” -> sentence too long, please rephrase into two sentences.

L361-365 “Hydrocarbon generation modeling using type I and II organic matter within non marine…” -> sentence too long, please rephrase into two sentences.

L377 express as a mathematical expression i.e. an equation (1) with corresponding notations and units.

385-386 Shorten title of Table 1.

L406 write an equation in a math program and add a notation for equation (2).

L432-435 “Previous research combined with the establishment of the sequence stratigraphic framework for the…”  -> sentence too long, please rephrase into two sentences.

L443-447 “Finally, oil shales buried at depths of…” -> sentence too long, please rephrase into two or more sentences.

L449 “Figure 16. Model showing the possible ICP extraction of oil from K2qn1member oil shales of the Songliao Basin; adapted from Zhao et al. (2013)18.” -> Figure 16. Model showing the possible ICP extraction of oil from K2qn1member oil shales of the Songliao Basin; adapted from [18]

L453-456 “The Songliao Basin contains three thick K2qn1 oil shale layers with the N1 oil shale layer in this member having a lower average thickness (2.1m) and quality (average TOC of 6.3wt.%) than the N2 (average oil shale thickness of 6.0m average TOC of 7.8 wt.%) and N3 (average oil shale thickness of 6.2m average TOC of 6.7wt.%) oil shale layers within this member.” -> The Songliao Basin contains three thick K2qn1 oil shale layers:

a) N1 oil shale layer in this member having a lower average thickness (2.1m) and quality (average TOC of 6.3wt.%)

b) N2 (average oil shale thickness of 6.0m average TOC of 7.8 wt.%),

c) N3 (average oil shale thickness of 6.2m average TOC of 6.7wt.%) oil shale layers within this member.

L475-478 “The hydrocarbon generating…”-> sentence too long, please rephrase into two sentences.

Kind regards,

Reviewer

Author Response

Answer to the question asked by the fourth reviewer:

Dear reviewer:

We have revised the manuscript according to the suggestions. Thank you for your detailed review of this article and large amount revision of specific contents. The following are the answers or explanations to the questions asked by you.

General Comments:

Question1: Incorrect citation throughout the text. It is correct e.g. “of oil generated during the middle part of this test test [1]”.

Answer---- The layout of references in this paper has been modified as required.

Question2: Check spaces throughout all paper (a lot of connected words in the text ...)..

Answer---- We have revised the words.

Question3: Write the individual contribution of each author in "Author Contributions".

Answer---- We have clearly described each person's contribution to the article, as follows: Penglin Zhang, Yinbo Xu and Qingtao Meng contributed to edit in this work, Jiaqiang Zhang and Yinbo Xu  provided the funding, Lin Shen and Shuaihua Zhang made the formal analysis and samples resources, Qingtao Meng and Zhaojun Liu made the validation.

Question4: The font size of the text in the title of the Figures throughout the paper should be adjusted in accordance with the "Instructions for Authors" (e.g. L-96; L-192…).

Answer---- The font size of the text in the title of the Figures throughout the paper have been revised.

 Question5: Insufficient cited literature of technology ICP in the "Introduction". Please cite relevant literature.

Answer---- We have added five references related to oil shale ICP in the "Introduction", mainly from Energies, AAPG and PETROL. EXPLOR. DEV., and layout in the relevant content.

Question6: Correct the references in the "References" in accordance with the "Instructions for Authors".

Answer---- We have revised the references based on "Instructions for Authors".

Question7: Too long "Abstract" needs to be shortened.

Answer---- We have revised and shorten the "Abstract", and highlight this article's new understanding of geochemistry, mineralogy and sequence stratigraphy related to ICP, as follows:

The Songliao Basin contains some of the largest volumes of oil shales in China, however, these energy sources are located in areas covered by arable land, meaning that the best way of exploiting them is likely to be environmentally friendly in situ conversion processing (ICP). Whether the oil shales of the Songliao Basin in the Qingshankou Formation are suitable for ICP remain controversial. In this paper, through sequence stratigraphic correlations, three main thick oil shale layers (N1,N2 and N3) of sequence1(Sq1) unit in the first member of Qingshankou Formation (K2qn1) are confirmed consistently present throughout the Southeastern Uplift region of the basin. The spectral trend attributes reflect lake reached a maximum flood surface of the K2qn1 in N2 oil shale layer, total organic carbon (TOC) and Fischer assay (FA) oil yield are significantly increasing. The N2 and N3 oil shale layers were deposited in a high lake level environment associated with ingressions of ocean water. Oil shale in these layers with the characteristics of high TOC(maximum of 23.9 wt.%;average of 7.2 wt.%), abundance of aquatic organic matter (OM)(maximum Hydrogen index(HI) of 1080.2 mg/g; average of 889.9 mg/g)  and carbonate contents(maximum of 29.5 %;average of 15.4 %).The N2 and N3 oil shale layers have higher brittleness index (BI) values (generally 40%–50%), larger cumulative thicknesses (maximum of 13.3m;average of 12.0m), and much higher source potential index(SPI)  values (0.92 and 0.88 tHC/m2, respectively) than the N1 oil shale layer within Sq1 transgressive system tracts (TST), indicating the N2 and N3 layers are prospective targets for ICP. In addition, oil shales buried to depths of <1000 m have strong hydrocarbon generation capacities that make them suitable for ICP.

Specific Comments:

Question8: L25-29 “The N2 and N3oil shale layers within the Sq1 unit were deposited in a high lake level environment associated with ingressions of ocean water, generating oil shales with high TOC(maximum of 23.9 wt.%;average of 7.2 wt.%),abundance of aquatic organic matter (OM)(maximum Hydrogen index(HI)of 1080.2 mg/g; average of 889.9 mg/g)  and carbonate contents(maximum of 29.5 %;average of 15.4 %).” -> sentence too long, please rephrase into two sentences.

Answer---- We have revised the sentence, as follows: 

The N2 and N3 oil shale layers were deposited in a high lake level environment associated with ingressions of ocean water. Oil shale in these layers with the characteristics of high TOC(maximum of 23.9 wt.%;average of 7.2 wt.%), abundance of aquatic organic matter (OM)(maximum Hydrogen index(HI) of 1080.2 mg/g; average of 889.9 mg/g)  and carbonate contents(maximum of 29.5 %;average of 15.4 %).

Question9: L37-38 Keywords:… -> without bold. 

Answer---- We have revised the Keywords.

 Question10: “Chinese oil shale resources are” -> Chinese oil shale reserves are…. 

Answer---- We have revised the word.

Question11: “billion tons of shale that can yield 61 billion tons of shale oil,” -> billion tons of oil 61 billion tons of shale oil, of which recoverable reserves are 61 billion tons of oil.

Answer---- We have revised the sentence.

Question12: L44 “, with the Upper Cretaceous of the…” -> Upper Cretaceous of the…

Answer---- We have revised the sentence.

Question13: L46 “north eastern China crops out over an area of ca 26x104 km2 and is covered by 559x104 hectares of…” -> north eastern China crops out over an area of 260 000 km2 and is covered over 5 590 000 hectares of…

Answer---- We have revised the sentence.

Question14: L47-50 “The basin records a rapid and large-scale lake transgression event during the deposition of…” -> sentence too long, please rephrase into two sentences.

Answer---- We have revised the sentence, as follows: 

The basin records a rapid and large-scale lake transgression event during the deposition of the first member of the Qingshankou Formation(K2qn1). Thick and dark shales and oil shales rich in organic matter, both of which are widely distributed throughout K2qn1 stratum in the basin and are thought to be ideal targets for oil extraction.

Question15: L55 “Research into the insitu conversion…” -> Research into ICP…

Answer---- We have revised the sentence.

Question16: L60 “University in cooperation with Israel”-> company name from Israel?

Answer---- We have revised the sentence. The name of company is Asia technology co., LTD.

Question17: L60 “, leading to a pilot test of the in situ conversion processing of oil” -> , leading to a pilot test of ICP of oil…

Answer---- We have revised the sentence.

Question18: L69-72 “However, since K2qn1 are dominated by dark…” -> sentence too long, please rephrase into two or more sentences.

Answer---- We have revised the sentence. as follows: 

However, since K2qn1 are dominated by dark mudstone, and traditional sequence division scheme is significantly influenced by subjective human factors. Sequence boundary of K2qn1 is controversial and poorly correlated, unable to be used to effectively predict the distribution of oil shales, and plays a limited role in guiding exploration and production.

Question19: L112-115 “. This approach is based on a cyclical stratigraphic theoretical basis and converts well…” -> sentence too long, please rephrase into two or more sentences.

Answer---- We have revised the sentence. as follows: 

This approach is based on a cyclical stratigraphic theoretical basis and converts well logs into a single integrated prediction error filter analysis (INPEFA) curve using modern digital signal processing techniques. INPEFA highlights the often hidden characteristics of stratigraphic cycles within well logs[17].

Question20: L118-120 “ However, significant changes in eustatic lake level during the deposition of the…” -> sentence too long, please rephrase into two sentences.

Answer---- We have revised the sentence. as follows: 

However, significant changes in eustatic lake level during the deposition of the K2qn1 member caused significant variations in the abundance of shale lithologies. Therefore the changes in the amplitude of the Gamma Ray(GR)curve are likely to indicate changes in sedimentary cycles.

Question21: L120 define "GR"

Answer---- We have defined "GR", as Gamma Ray.

Question22: L120-124 “In…” -> sentence too long, please rephrase into two sentences.

Answer---- We have revised the sentence, as follows: 

In addition, the fact that the GR curve is only slightly affected by borehole conditions, and is already included within the logging suites of existing wells in this area.Thus, these data were converted to yield INPEFA curves that, in turn, were used to define the sequence stratigraphy using the following approaches.

Question23: L130-132 “stratigraphic continuity where negative peaks ( i.e., large negative error s ) represent possible flooding surfaces , positive peaks (large positive error s ) represent possible sequence boundaries , and different peaks (different errors) are…” -> stratigraphic continuity: negative peaks represent possible flooding surfaces , positive peaks represent possible sequence boundaries, and different peaks  are…

Answer---- We have revised the sentence.

Question24: delete the point at the beginning.

Answer---- We have revised the sentence.

Question25: L261 delete point at the end of the subtitle

Answer---- We have revised the subtitle.

Question26: L246-249 “The average ‘‘true’’ HI values for the N3…” -> sentence too long, please rephrase into two or more sentences.

Answer---- We have revised the sentence, as follows: 

The average ‘‘true’’ HI values for the N3 and N2 oil shale layers are 1105 and 959 mg HC/gTOC, respectively (r2 = 0.92 for N1 and 0.98 for N2). However the N1 oil shale layer yielded a lower ‘‘true’’ HI value of 760 mg HC/g TOC, and with a poorer correlation between the S2 and TOC values for these samples(r2 = 0.56).

Question27: L261 delete point at the end of the subtitle

Answer---- We have revised the subtitle.

Question28: L265-269 “Comparing the GR curves…” -> sentence too long, please rephrase into two or more sentences.

Answer---- We have revised the sentence, as follows: 

Comparing the GR curves with this framework indicates that the spectral trends of these curves correlate well with variations in the stratigraphy and sedimentary cyclicity of the K2qn1 member in the Southeastern Uplift zone. In K2qn1, boundaries of the three third-order sequences and their internal constitutive characteristics clearly evident in Figure 8.

Question29: L278-282 “in the form of 1) a rapid decrease in GR curve 278 data , 2) low wavelength but high amplitude MESA data where amplitude peaks occur at 100 ka intervals, indicating that these short term cycles are predominantly controlled by eccentricity variationsvariations11 3) negative peaks in PEFA curve data, and 4) INPEFA curve values close to one that represent a negative turning point from positive to negative trends” -> Rearrange text: in the form:

1) a rapid decrease in GR curve 278 data,

2) low wavelength but high amplitude MESA data where amplitude peaks occur at 100 ka intervals, indicating that these short term cycles are predominantly controlled by eccentricity variationsvariations11

3) negative peaks in PEFA curve data,

4) INPEFA curve values close to one that represent a negative turning point from positive to negative trends.

Answer---- We have revised the sentence.

Question30: L282-285 “The TST are dominated by dark…” -> sentence too long, please rephrase into two sentences.

Answer---- We have revised the sentence, as follows: 

The TST are dominated by dark gray to gray–black mudstones and oil shales, all of which become darker and contain more carbonate minerals from the base to the top of these tracts (Figure 9). Oil shales in this unit with the characteristic of bioclastic bands (Figure 9),

Question31: L315 In Figure 10a, add the corresponding equation y =kx+l.

Answer---- We have revised the Figure 10a.

Question32: L319-324 “The majority of the quartz within the oil shale samples from the three…” -> sentence too long, please rephrase into two or more sentences.

Answer---- We have revised the sentence.

Question33: L326-329 “This suggests that the periods of N2 and N3 deposition were…” -> sentence too long, please rephrase into two sentences.

Answer---- We have revised the sentence.

Question34: L361-365 “Hydrocarbon generation modeling using type I and II organic matter within non marine…” -> sentence too long, please rephrase into two sentences.

Answer---- We have revised the sentence, as follows: 

Hydrocarbon generation modeling using type-I and -II organic matter within non-marine shales indicates that vitrinite reflectance (Ro) values of 0.5% are associated with OM that has 90% of the original hydrocarbon generation potential. However, oil shale with Ro values of >1.0% are associated with a rapid decline in hydrocarbon generation potential.

Question35: L377 express as a mathematical expression i.e. an equation (1) with corresponding notations and units.

Answer---- We have revised the sentence, as follows: 

This can be further quantified using the source potential index(SPI)[37].

SPI=( S1+S2)´ h´ p/1000-----------(1)

Where h (m)is thickness and p (t/m3) is bulk density.

Question36: 385-386 Shorten title of Table 1.

Answer---- We have revised title of Table 1.

Question37: L406 write an equation in a math program and add a notation for equation (2).

Answer---- We have revised equation.

Question38: L432-435 “Previous research combined with the establishment of the sequence stratigraphic framework for the…”  -> sentence too long, please rephrase into two sentences..

Answer---- We have revised the sentence, as follows: 

The sequence stratigraphic framework for the Qingshankou Formation based on spectral trend attribute analysis indicates that oil shales within this formation have a well-defined vertical distribution pattern. Thus oil shale layers deposition was controlled by the sequence stratigraphic framework of this area (Figure16).

Question39: L443-447 “Finally, oil shales buried at depths of…” -> sentence too long, please rephrase into two or more sentences.

Answer---- We have revised the sentence.

Question40: L449 “Figure 16. Model showing the possible ICP extraction of oil from K2qn1member oil shales of the Songliao Basin; adapted from Zhao et al. (2013)18.” -> Figure 16. Model showing the possible ICP extraction of oil from K2qn1member oil shales of the Songliao Basin; adapted from [18]

Answer---- We have revised the title.

Question41: L453-456 “The Songliao Basin contains three thick K2qn1 oil shale layers with the N1 oil shale layer in this member having a lower average thickness (2.1m) and quality (average TOC of 6.3wt.%) than the N2 (average oil shale thickness of 6.0m average TOC of 7.8 wt.%) and N3 (average oil shale thickness of 6.2m average TOC of 6.7wt.%) oil shale layers within this member.” -> The Songliao Basin contains three thick K2qn1 oil shale layers:

  1. a) N1 oil shale layer in this member having a lower average thickness (2.1m) and quality (average TOC of 6.3wt.%)

  1. b) N2 (average oil shale thickness of 6.0m average TOC of 7.8 wt.%),

  1. c) N3 (average oil shale thickness of 6.2m average TOC of 6.7wt.%) oil shale layers within this member.

Answer---- We have revised the sentence by your comments.

Question42: L475-478 “The hydrocarbon generating…”-> sentence too long, please rephrase into two sentences.

Answer---- We have revised the sentence, as follows: 

The hydrocarbon generating capacity of the oil shales also indicates that oil shale layers buried to depths of<1000 m are suitable for ICP. Sequence stratigraphy for oil shale layers in the Southeastern Uplift of Songliao Basin indicating that the ICP-amenable N2 and N3 layers are widespread and have a stable distribution throughout this region.

Thank you again for your suggestions, which were all very helpful for improving our paper. We also sincerely appreciate Energies and you for giving us the opportunity to revise our manuscript.

Round 2

Reviewer 4 Report

Dear authors,

The corrections I requested for the paper "Sequence stratigraphy and geochemistry of oil shale deposits in the Upper Cretaceous Qingshankou Formation of the Songliao Basin, NE China: implications for the geological optimization of insitu oil shale conversion processing" were successfully made. The quality and clarity of the text and results has been significantly improved. The scientific contribution is visible and applicable and, following the proposed corrections, future research on this topic can be compared and developed. I wish successful further research.

Best regards,

Reviewer